# *Soft Reasoning*: Navigating Solution Spaces in Large Language Models through Controlled Embedding Exploration

Qinglin Zhu [1] [*]   Runcong Zhao [1] [*]   Hanqi Yan [1]   Yulan He [1] [2]   Yudong Chen [3]   Lin Gui [1]

## Abstract

Large Language Models (LLMs) struggle with complex reasoning due to limited diversity and inefficient search. We propose *Soft Reasoning*, an embedding-based search framework that optimises the embedding of the first token to guide generation. It combines (1) embedding perturbation for controlled exploration and (2) Bayesian optimisation to refine embeddings via a verifier-guided objective, balancing exploration and exploitation. This approach improves reasoning accuracy and coherence while avoiding reliance on heuristic search. Experiments demonstrate superior correctness with minimal computation, making it a scalable, model-agnostic solution.

## 1. Introduction

Large language models (LLMs) have demonstrated remarkable potential in various reasoning tasks, particularly on relatively simple and common benchmarks (Huang & Chang, 2023; Xu et al., 2025). Despite this, they still face significant limitations in complex tasks (Lightman et al., 2024; Wang et al., 2023a), which often require deeper levels of thought, and answers generated solely based on maximum likelihood are frequently incorrect. To increase the probability that the correct answer is included among generated candidates, many existing approaches aim to enhance generation diversity through multiple sampling (Lightman et al., 2024). A common mechanism for achieving such diversity is temperature scaling, which adjusts the randomness of token selection (Brown et al., 2024). Complementary to this, planning-based methods, such as chain-of-thought reasoning (Wei et al., 2022; Wang et al., 2023a) or tree-structured search (Yao et al., 2023), attempt to locate the

---

[*]Equal contribution [1]King's College London, UK [2]The Alan Turing Institute, UK [3]University of Warwick, UK. Correspondence to: Yudong Chen <yudong.chen@warwick.ac.uk>, Lin Gui <lin.1.gui@kcl.ac.uk>.

*Proceedings of the 42$^{nd}$ International Conference on Machine Learning*, Vancouver, Canada. PMLR 267, 2025. Copyright 2025 by the author(s).

correct answer by following language-based instructions.

Despite these efforts, two key challenges remain: (1) Enhancing generation diversity typically relies on increasing the temperature parameter, which flattens the token distribution. This, however, does not necessarily result in better coverage of the correct answer, as increasing low-probability token likelihood indiscriminately may introduce noise rather than meaningful exploration (Holtzman et al., 2020). (2) Existing planning and search methods such as sampling multiple reasoning paths rely heavily on heuristic strategies, guided by prompts (Hao et al., 2023; Qi et al., 2025b). However, these approaches do not directly adjust for the model's internal representations, thereby making the search process inefficient and highly dependent on surface-level prompt variations. This often leads to a "wild-goose chase", where search remains constrained by randomness and indirect heuristics rather than systematic optimisation.

To address these challenges, we propose *Soft Reasoning*, a novel approach using controlled embedding exploration: (1) By injecting a Gaussian embedding into the decoding of the first answer token, we can adjust the distribution of low-probability tokens in a more controlled manner than uniform temperature tuning, leading to more flexible generation. (2) Treating the LLM as a black box verifier, we apply Bayesian optimisation (Frazier, 2018) on the injected embedding to maximise a verification-based reward. This allows us to use observerd rewards to directly guide the exploration in the embedding space. As a result, *Soft Reasoning* improves performance without a strong verifier—even when both generation and verification originate from the same model.

As illustrated in Figure 1, *Soft Reasoning* leverages injected vectors to change the distribution of the next generated token, rather than simply flattening the output probability curve. In this injection-based generation process, decoding is performed via greedy search, ensuring that each injected vector corresponds to a unique generated sequence. This guarantees both controllability and repeatability. The effect of each injected vector can then be evaluated using a reward function that accounts for both correctness and coherence of the generated sequence. Next, in a sequential way, we identify promising directions for further exploration based on all observed injection-reward pairs by utilising Bayesian

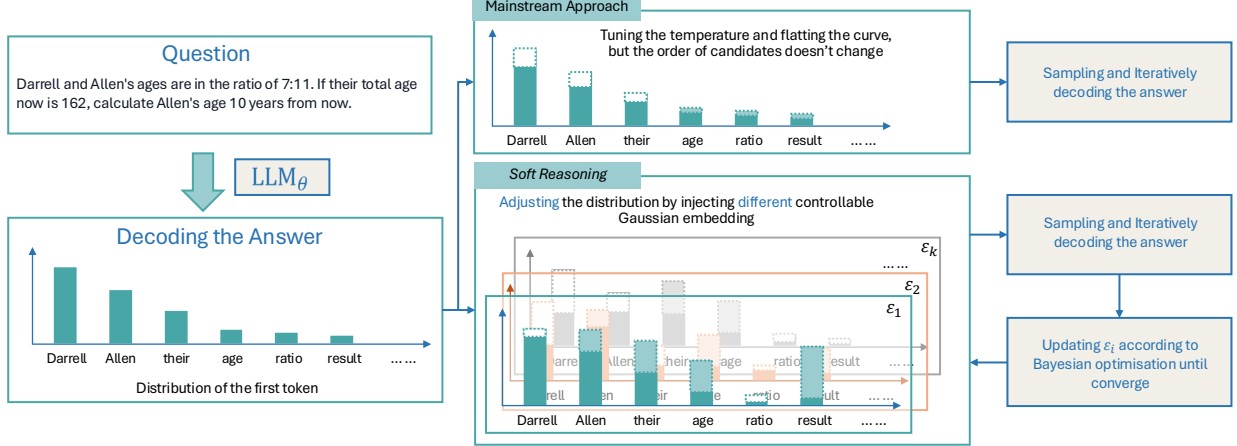

*Figure 1.* Comparison of Mainstream and Proposed Approaches.

optimisation. Again, thanks to the one-to-one mapping between initial vectors and generation outcomes, this process enables an effective search for optimal vectors that enhance generation quality. A notable advantage of this approach is that it operates without requiring access to the internal parameters of the LLM, allowing for efficient, low-resource control over generation behaviour.

Our contributions can be summarised as follows:

- We propose a reasoning method, *Soft Reasoning*, which combines embedding perturbation and Bayesian optimisation to better control low-probability token selection, enabling more flexible and diverse generation than temperature tuning.
- Instead of language-based instruction or heuristic search, our method directly optimises the embedding of the first generated token to control the direction of thinking and exploration in LLM, effectively reducing the searching and reasoning complexity, and improving efficiency and accuracy.
- *Soft Reasoning* is able to control and optimise the reasoning without accessing model parameters or requiring additional verifier, allowing seamless integration into different mainstream LLMs. Experiments across a variety of LLMs and reasoning tasks demonstrate improved correctness and efficiency of *Soft Reasoning* over traditional decoding.

## 2. Related Work

### 2.1. Decoding Strategies and Diversity

Recent advances in LLM decoding aim to enhance diversity for tasks requiring creativity and exploration. Traditional methods such as greedy and beam search often produce repetitive outputs (Holtzman et al., 2020; Welleck et al., 2019), while sampling-based approaches (top-$k$, nu-

cleus) introduce randomness but struggle to balance quality and diversity (Fan et al., 2018; Holtzman et al., 2020). High-temperature settings can lead to incoherent outputs (Minh et al., 2025), and adaptive methods like min-$p$ sampling (Minh et al., 2025) require careful tuning. Debiasing-Diversifying Decoding (D3) mitigates amplification bias but increases computational cost (Bao et al., 2024). Crucially, most methods overlook the impact of initial token selection, which significantly influences reasoning outcomes (Wang & Zhou, 2024). Our approach addresses this by perturbing initial token embeddings with Gaussian noise, reshaping the probability distribution to improve exploration while maintaining quality and efficiency.

### 2.2. Efficient Exploration of Solution Spaces

Efficient solution space exploration is crucial for enhancing LLM reasoning while maintaining practical computational costs. Increasing generated samples improves coverage (Brown et al., 2024) but is computationally prohibitive. Optimising test-time compute allocation is more effective than scaling model size (Snell et al., 2025), though it requires task-specific strategies. Mutual reasoning frameworks leveraging self-play and MCTS (Qi et al., 2025b; Yan et al., 2024), as well as Tree of Thoughts (ToT) (Yao et al., 2023), explore multiple reasoning paths but incur high computational overhead. Thought Space Explorer (TSE) (Zhang & Liu, 2024) enhances reasoning breadth but at additional cost. *Soft Reasoning* refines these approaches by integrating controlled initial-token embedding perturbations with a strategic search algorithm inspired by MCTS and mutual reasoning. By introducing exploration early through embedding perturbation and guiding search via a verifier, we improve efficiency without excessive computational overhead, striking a balance between exploration and exploitation to optimise reasoning performance.

## 3. Preliminary: Temperature Scaling

A common approach for generating diverse outputs is temperature scaling, which controls the randomness in the token generation process by modifying the softmax distribution over the model's output logits. For a given temperature $\tau > 0$, the probability of selecting token $w^{(t)}$ at time step $t$ is given by:

$$P(w^{(t)} \mid w^{(1:t-1)}; \theta, \tau) = \frac{\exp(\ell_{t,w^{(t)}}/\tau)}{\sum_w \exp(\ell_{t,w}/\tau)},$$

where $w^{(1:t-1)}$ represents the sequence of tokens $\{w^{(1)}, \ldots, w^{(t-1)}\}$ generated from the first token up to the $(t-1)^{th}$ token, $\ell_{t,w}$ denotes the logit at time $t$ corresponding to token $w$, and $\tau$ controls the sharpness of the distribution. This scaling flattens the distribution but preserves the relative ranking of token probabilities. When $\tau$ is low, the results concentrate on a few high-probability tokens, leading to overly deterministic generations with limited diversity. When $\tau$ is high, the model may sample low-probability tokens, leading to incoherent outputs.

While this approach increases diversity, it lacks control, blindly flattening token probabilities; adaptability, as it ignores verifier feedback; and efficiency, often requiring multiple samples or retraining (Joy et al., 2023; Xie et al., 2024). These limitations make it ineffective for structured reasoning tasks that demand precise and efficient exploration.

Generating accurate answers in complex tasks requires both exploring reasoning paths and verifying for their correctness. To achieve this, we propose a two-step framework as shown in Figure 2: (1) Embedding perturbation applies a Gaussian adjustment to the first-token embedding for controlled modifications beyond uniform tuning, (2) Bayesian optimisation refines the perturbed embedding to maximise a verifier-guided reward, improving reasoning path selection.

### 3.1. Embedding Perturbation

Given a generative model $g_\theta$ and a natural language question prompt $q$, the first token $w^{(1)}$ is generated using greedy decoding, which selects the token with the highest probability from the model's predicted distribution:

$$w^{(1)} = \underset{w}{\arg\max} \, P(w \mid q; \theta),$$

where $P(\cdot \mid q; \theta)$ represents the probability distribution over the possible tokens predicted by the model $g_\theta$, with input $q$.

Let $z \in \mathbb{R}^D$ represent the embedding of the token $w^{(1)}$. This embedding serves as a prior, representing a "correct starting point" in the latent space. To explore the neighbourhood of this embedding, we define a set of perturbed embeddings $x_i$ for $i = 1, \ldots, k$ as follows:

$$x_i = z + \sigma \varepsilon_i, \quad \varepsilon_i \sim \mathcal{N}(0, I),$$

where $\varepsilon_i$ represents independent random perturbations drawn from a standard normal distribution, and $\sigma$ is a scaling factor controlling the magnitude of the perturbation. This formulation allows us to sample from the local vicinity of the original embedding $z$, exploring variations around the initial token representation.

For each perturbed embedding $x_i$, we introduce a corresponding special token mapped to $x_i$ and add it to the vocabulary. This special token is then used as the first token for generating an answer. Since we use **greedy decoding**, this $x_i$ **fully determines** the entire output sequence, i.e. the remaining tokens $w_i^{(2)}, w_i^{(3)}, \ldots, w_i^{(L)}$ are then deterministically generated in a sequential manner:

$$w_i^{(t)} = \underset{w}{\arg\max} \, P(w \mid x_i, w_i^{(2:t-1)}, q; \theta).$$

We denote $y_i := w_i^{(1:L)}$ to be the complete output based on the initial perturbed embedding $x_i$. We then repeat this process $k$ times to generate $k$ different answers: $y_1, \ldots, y_k$. Since each output $y_i$ is fully determined by $x_i$, embedding perturbation effectively serves as a sampling mechanism over the entire answer space.

### 3.2. Exploring the Embedding Space

Randomly sampling points with infinite computational resources could theoretically approximate the optimal solution, but this approach is highly inefficient, especially given the computational expense of sampling with an LLM. Instead, we adopt Bayesian optimisation, which consists of two key components: an *objective function* and an *acquisition function* that determine where to sample next. We use Expected Improvement (EI) as our acquisition function, which offers a closed-form solution (Frazier, 2018) with negligible computational cost, making it significantly more efficient by comparison. EI effectively balances exploration (searching uncertain regions) and exploitation (refining promising areas), selecting the point with the highest EI at each iteration to guide the optimisation process toward convergence.

**Optimisation Objective.** To evaluate the objective function with $k$ sampled perturbed embeddings, we consider the sequence $x_{1:k} = \{x_1, \ldots, x_k\}$, where each $x_i \in \mathbb{R}^D$. The corresponding answers are then generated as described above: $y_{1:k} = \{y_1, \ldots, y_k\}$. Comparing and refining multiple generated answers has been shown to improve performance (Miao et al., 2024). Additionally, since LLMs are primarily trained for text generation rather than explicit judgment, prompting them to regenerate and compare outputs can yield better results (Zhang et al., 2024). Building on these insights, we propose a *verifier-guided approach*, where the model evaluates a batch of candidate answers and produces a refined output $y_v = \mathcal{V}(y_{1:k})$. The correctness of

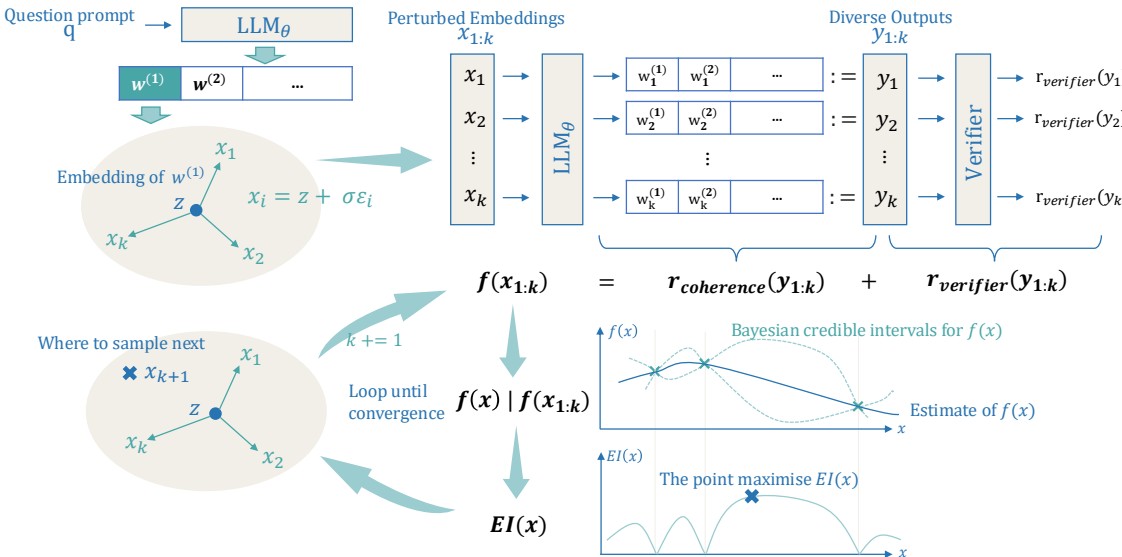

*Figure 2.* **Overview of *Soft Reasoning*.** Starting with a natural language question prompt, the model generates initial token embeddings $w^{(1)}$, which, due to greedy decoding, determine the entire output. These embeddings are perturbed to create candidate embeddings $x_{1:k}$, leading to outputs $y_{1:k}$ through greedy search, which are then evaluated for coherence and verifier feedback. A Bayesian optimisation framework updates its estimation of the space based on this feedback and selects the next sampling point that maximises the expected improvement, balancing exploration and exploitation to refine the search for high-quality outputs.

an answer $y$ is then assessed as a binary indicator (0 or 1), based on its alignment with the verifier's final output. The verifier is the same model as the generator employed.

The embedding space may not be uniform, implying that perturbations in different directions can lead to *uneven semantic shifts* (Li et al., 2023; Park et al., 2024). In some dimensions, even small perturbations can significantly alter meaning, potentially disrupting grammar or context consistency and leading to incoherent outputs. To address this issue, we introduce the coherence term to prune low-quality generations, ensuring that only outputs with desirable semantic and syntactic properties are retained. To evaluate the quality of a generated output $y$, we define an objective function $f(x)$ that balances correctness and fluency:

$$f(x) = r_{\text{verifier}}(y) + r_{\text{coherence}}(y), \quad (1)$$

where:

- **Verifier Score ($r_{\text{verifier}}$):** This is a binary indicator provided by the verifier, reflecting the correctness of $y$:

$$r_{\text{verifier}}(y) = \mathbb{1}_{\{y_v = y\}};$$

- **Coherence ($r_{\text{coherence}}$):** This term evaluates the fluency of the generated sequence based on token probabilities:

$$r_{\text{coherence}}(y) = \sum_{i=1}^{T} \log P(w^{(i)}),$$

where $P(w^{(i)})$ is the probability of generating token $w^{(i)}$ from the LLM's entire vocabulary.

**Bayesian Optimisation.** Our goal is to maximise $f(\cdot)$, as defined in (1), over the embedding space $\mathbb{R}^D$. To optimise this black-box function, Bayesian optimisation uses a prior distribution on the domain to represent our beliefs about the behavior of the function and iteratively updates this prior using newly acquired data. Specifically, we model the prior joint distribution as a multivariate Gaussian distribution:

$$f(x_{1:n}) \sim \mathcal{N}\big(\mu_0(x_{1:n}), \Sigma_0(x_{1:n}, x_{1:n})\big),$$

where $\mu_0(x_{1:n})$ is the prior mean vector, and $\Sigma_0(x_{1:n}, x_{1:n})$ is the prior covariance matrix.

After observing $f(x_{1:k})$, we aim to infer the value of $f(x)$ at a new point $x$. Using Bayes' rule (Rasmussen & Williams, 2006), we update the posterior distribution of $f(x)$ conditioned on these observed values:

$$f(x) \mid f(x_{1:k}) \sim \mathcal{N}(\mu_k(x), \sigma_k^2(x)). \quad (2)$$

Here, $\mu_k(x)$ and $\sigma_k^2(x)$ represent the posterior mean and variance, respectively. A detailed discussion on the choice of the prior distribution and the computation of the posterior distribution is provided in Appendix A.2.

A naive way to find the maximiser at this stage would be to select among the previously evaluated points $x_1, \ldots, x_k$ the one with the highest observed function value. Let $f_k^* := \max_{m \le k} f(x_m)$ denote this value. If we were to sample another point $x \in \mathbb{R}^D$ and observe $f(x)$, then the value of the best observed point would either be $f(x)$ (if $f(x) \ge f_k^*$) or $f_k^*$ (if $f(x) < f_k^*$). The improvement in the value of the

best observed point could be expressed as $[f(x) - f_k^*]^+ := \max(f(x) - f_k^*, 0)$.

While we would ideally choose $x$ to maximise this improvement, $f(x)$ is unknown until after the evaluation. Instead, we select $x$ that maximises the expected improvement under the posterior distribution, defined as

$$\text{EI}_k(x) := \mathbb{E}_k\big[[f(x) - f_k^*]^+\big], \tag{3}$$

where $\mathbb{E}_k$ denotes the expectation taken with respect to the posterior distribution (2). Using integration by parts, we can write EI (3) in a closed-form expression:

$$\text{EI}_k(x) = \big[\mu_k(x) - f_k^*\big]^+ + \sigma_k(x)\phi\bigg(\frac{\mu_k(x) - f_k^*}{\sigma_k(x)}\bigg)$$
$$- \big|\mu_k(x) - f_k^*\big|\Phi\bigg(\frac{\mu_k(x) - f_k^*}{\sigma_k(x)}\bigg),$$

where $\phi$ and $\Phi$ denote the probability density function and the cumulative distribution function of the standard normal distribution, respectively.

Our next sampling point, $x_{k+1} \in \mathbb{R}^D$, is the maximiser of EI. We then iteratively update the posterior distribution and the EI function. Details on how we select $x$ to maximise $\text{EI}_k(x)$ can be found in Appendix A.3. Convergence is considered achieved when the change in the objective function between consecutive iterations satisfies $|f_k - f_{k-1}| < \epsilon$, where $\epsilon$ is a predefined threshold. Additionally, the algorithm terminates after a maximum of $K$ iterations if convergence has not been reached.

In defining $f(x)$, we assume an ideal verifier with perfect accuracy, meaning it provides an error-free assessment of correctness. However, in practice, the verifier's accuracy is less than 1, introducing uncertainty into its evaluations. To address this noise in Bayesian optimisation, we use an adaptive version of the EI acquisition function that explicitly incorporates observation uncertainty. This adaptation dynamically adjusts the exploration rate based on uncertainty, ensuring a higher probability of convergence while balancing exploration and exploitation (Vakili et al., 2021; Tran-The et al., 2022). Theoretical foundations and implementation details are provided in Appendix A.4.

**Dimension Reduction.** One shortcoming of using traditional Bayesian optimisation methods for identifying the point with maximum EI (Mockus, 1975; Hvarfner et al., 2024) is that they perform poorly when the search space exceeds 20–30 dimensions due to the curse of dimensionality (Kandasamy et al., 2015; Letham et al., 2020; Wang et al., 2023b). In high-dimensional spaces, surrogate models require an exponentially larger number of points to accurately estimate the maximum of the EI function, making optimisation highly inefficient. With the dimension of embedding vectors for LLMs typically ranging from 768 to 8192 or more, traditional methods are impractical in our setting.

To address this, we leverage a dimension reduction approach based on random embeddings (Wang et al., 2016; Nayebi et al., 2019). Specifically, if a function $f : \mathbb{R}^D \to \mathbb{R}$ has an effective dimension $d_e \leq D$, then with high probability, there exists a lower-dimensional representation $g(u) := f(Au)$, where $A$ is a random projection matrix. This allows optimisation to be performed in a lower-dimensional space $\mathbb{R}^d$ instead of the original $\mathbb{R}^D$. Using this approach, we iteratively optimise the function in the reduced space and map solutions back to the original space. Theoretical foundations and implementation details are provided in Appendix A.5.

## 4. Experiments

We benchmark *Soft Reasoning* against strong baselines and conduct ablation studies.

### 4.1. Experimental Setup

**Datasets and Models.** We conduct experiments using three LLMs: Llama-3.1-8B-Instruct (Meta, 2024), Qwen2-7B-Instruct, Qwen2-70B-Instruct (Yang et al., 2024), and Mistral-8B-Instruct (Jiang et al., 2023). The models are evaluated on four benchmark datasets, including three complex mathematical tasks (GSM8K (Cobbe et al., 2021), GSM-Hard (Gao et al., 2023), SVAMP (Patel et al., 2021)), and one commonsense reasoning task StrategyQA (Geva et al., 2021). For the Qwen2-70B-Ins model, we additionally evaluate its performance on the AIME-2024 benchmark.

**Baselines.** Our baselines include: (1) CoT Prompting, which includes zero-shot CoT (Kojima et al., 2022) and few-shot CoT (Wei et al., 2022); (2) Self-Consistency (SC) Decoding (Wang et al., 2023c), which involves sampling answers at various temperatures $\tau \in \{0.4, 0.6, 0.8\}$ and selecting the final answer through majority voting; (3) FIRE (Chen et al., 2025), which adjusts the decoding process by setting the temperature of the first token to 30 to enhance diversity, while subsequent tokens are generated using the standard temperature setting; (4) CoT-Decoding (Wang & Zhou, 2024), which generates $k$ answers by sampling the top-$k$ tokens from the probability distribution of the first token. Each of these top-$k$ tokens is used as the starting point for decoding the remainder of the answer; and (5) RAP (Hao et al., 2023), which uses Monte Carlo Tree Search to explore reasoning paths strategically, balancing exploration and exploitation to find solutions efficiently. Note that RAP requires problem decomposition via examples; hence we only report its performance in the few-shot setting. Additionally, we compare *Soft Reasoning* with recent controlled generation approaches, including Trainable Prefix Scorers (Mudgal et al., 2024) and Constrained Fine-tuning (Qi et al., 2025a), with further details provided in Appendix B.2.

*Table 1.* Performances of different reasoning methods on Accuracy (%) across all benchmarks.

| Model | Method | GSM8K | | GSM-Hard | | SVAMP | | StrategyQA | |
|---|---|---|---|---|---|---|---|---|---|
| | | Zero Shot | Few Shot | Zero Shot | Few Shot | Zero Shot | Few Shot | Zero Shot | Few Shot |
| LLaMA3-8B-Ins | CoT | $53.0_{\pm0.0}$ | $77.4_{\pm0.0}$ | $14.0_{\pm0.0}$ | $28.0_{\pm0.0}$ | $61.0_{\pm0.0}$ | $83.0_{\pm0.0}$ | $58.5_{\pm0.0}$ | $68.5_{\pm0.0}$ |
| | SC($\tau=0.4$) | $73.0_{\pm1.6}$ | $80.4_{\pm1.4}$ | $25.7_{\pm0.4}$ | $31.8_{\pm1.8}$ | $79.1_{\pm1.2}$ | $87.1_{\pm1.0}$ | $64.7_{\pm0.7}$ | $71.6_{\pm0.8}$ |
| | SC($\tau=0.6$) | $73.6_{\pm2.5}$ | $80.6_{\pm1.5}$ | $24.5_{\pm1.1}$ | $31.2_{\pm1.3}$ | $76.1_{\pm3.9}$ | $87.7_{\pm1.2}$ | $59.9_{\pm2.0}$ | $71.3_{\pm1.5}$ |
| | SC($\tau=0.8$) | $65.0_{\pm2.0}$ | $81.1_{\pm1.1}$ | $21.8_{\pm1.3}$ | $30.8_{\pm0.9}$ | $69.6_{\pm2.0}$ | $87.4_{\pm1.2}$ | $54.4_{\pm2.6}$ | $72.7_{\pm1.2}$ |
| | FIRE | $73.8_{\pm2.3}$ | $79.6_{\pm2.9}$ | $25.2_{\pm3.0}$ | $25.7_{\pm2.1}$ | $81.5_{\pm0.8}$ | $87.6_{\pm2.0}$ | $63.0_{\pm3.7}$ | $72.8_{\pm1.5}$ |
| | CoT-Decoding | $73.9_{\pm1.9}$ | $80.3_{\pm1.7}$ | $24.8_{\pm1.3}$ | $30.3_{\pm1.3}$ | $83.2_{\pm1.2}$ | $88.2_{\pm1.0}$ | $64.6_{\pm1.6}$ | $73.3_{\pm1.8}$ |
| | RAP | - | $80.7_{\pm1.4}$ | - | $32.7_{\pm1.2}$ | - | $87.9_{\pm1.1}$ | - | $73.4_{\pm1.1}$ |
| | *Soft Reasoning* | $\mathbf{79.4}_{\pm1.2}$ | $\mathbf{84.3}_{\pm1.4}$ | $\mathbf{28.2}_{\pm1.8}$ | $\mathbf{35.7}_{\pm1.0}$ | $\mathbf{88.2}_{\pm1.3}$ | $\mathbf{90.2}_{\pm0.6}$ | $\mathbf{67.2}_{\pm0.7}$ | $\mathbf{75.6}_{\pm0.8}$ |
| | **w/o** $r_{\text{verifier}}$ | $76.8_{\pm1.0}$ | $82.0_{\pm0.5}$ | $26.3_{\pm1.3}$ | $34.8_{\pm0.3}$ | $86.7_{\pm1.2}$ | $89.5_{\pm0.5}$ | $66.2_{\pm2.8}$ | $74.3_{\pm1.6}$ |
| | **w/o** $r_{\text{coherence}}$ | $77.4_{\pm2.1}$ | $83.4_{\pm0.7}$ | $27.9_{\pm1.5}$ | $35.3_{\pm1.3}$ | $84.6_{\pm2.4}$ | $90.1_{\pm0.9}$ | $66.0_{\pm1.3}$ | $75.0_{\pm1.5}$ |
| Qwen2-7B-Ins | CoT | $64.5_{\pm0.0}$ | $82.5_{\pm0.0}$ | $40.0_{\pm0.0}$ | $55.5_{\pm0.0}$ | $43.5_{\pm0.0}$ | $86.0_{\pm0.0}$ | $63.0_{\pm0.0}$ | $70.0_{\pm0.0}$ |
| | SC($\tau=0.4$) | $81.2_{\pm0.6}$ | $85.7_{\pm1.5}$ | $47.5_{\pm1.4}$ | $55.4_{\pm0.7}$ | $72.3_{\pm2.0}$ | $90.3_{\pm1.2}$ | $67.1_{\pm1.5}$ | $71.1_{\pm1.6}$ |
| | SC($\tau=0.6$) | $80.2_{\pm1.9}$ | $85.4_{\pm0.9}$ | $46.2_{\pm1.9}$ | $53.4_{\pm0.6}$ | $77.3_{\pm1.2}$ | $90.4_{\pm0.6}$ | $67.5_{\pm0.7}$ | $69.1_{\pm1.2}$ |
| | SC($\tau=0.8$) | $80.0_{\pm0.9}$ | $85.1_{\pm1.6}$ | $47.3_{\pm1.3}$ | $55.4_{\pm0.9}$ | $78.6_{\pm2.1}$ | $90.6_{\pm1.2}$ | $67.0_{\pm1.0}$ | $70.1_{\pm0.8}$ |
| | FIRE | $81.0_{\pm1.8}$ | $83.0_{\pm1.3}$ | $45.1_{\pm2.0}$ | $51.0_{\pm1.8}$ | $76.3_{\pm2.2}$ | $90.6_{\pm0.2}$ | $67.6_{\pm0.8}$ | $68.1_{\pm0.8}$ |
| | CoT-Decoding | $82.0_{\pm2.8}$ | $84.5_{\pm2.1}$ | $46.7_{\pm2.3}$ | $52.1_{\pm1.0}$ | $78.6_{\pm1.6}$ | $89.7_{\pm0.5}$ | $65.9_{\pm1.5}$ | $69.5_{\pm2.1}$ |
| | RAP | - | $86.2_{\pm1.2}$ | - | $56.2_{\pm0.8}$ | - | $90.8_{\pm1.1}$ | - | $\mathbf{71.3}_{\pm1.3}$ |
| | *Soft Reasoning* | $\mathbf{88.6}_{\pm1.2}$ | $\mathbf{90.0}_{\pm1.4}$ | $\mathbf{53.7}_{\pm1.6}$ | $\mathbf{58.7}_{\pm0.5}$ | $\mathbf{83.4}_{\pm2.4}$ | $\mathbf{92.2}_{\pm0.8}$ | $\mathbf{68.1}_{\pm1.5}$ | $70.3_{\pm1.3}$ |
| | **w/o** $r_{\text{verifier}}$ | $87.0_{\pm1.0}$ | $89.7_{\pm1.8}$ | $51.2_{\pm2.1}$ | $58.3_{\pm1.0}$ | $73.5_{\pm4.0}$ | $90.5_{\pm1.5}$ | $66.0_{\pm0.9}$ | $68.3_{\pm1.6}$ |
| | **w/o** $r_{\text{coherence}}$ | $87.3_{\pm2.0}$ | $89.2_{\pm1.3}$ | $52.0_{\pm1.5}$ | $60.0_{\pm1.0}$ | $76.7_{\pm1.4}$ | $90.7_{\pm0.6}$ | $66.5_{\pm0.5}$ | $69.5_{\pm1.5}$ |
| Mistral-7B-Ins | CoT | $42.0_{\pm0.0}$ | $54.0_{\pm0.0}$ | $14.5_{\pm0.0}$ | $24.0_{\pm0.0}$ | $52.0_{\pm0.0}$ | $72.0_{\pm0.0}$ | $62.0_{\pm0.0}$ | $69.0_{\pm0.0}$ |
| | SC($\tau=0.4$) | $52.9_{\pm0.5}$ | $58.3_{\pm1.5}$ | $19.5_{\pm1.0}$ | $26.1_{\pm1.5}$ | $67.4_{\pm2.5}$ | $77.8_{\pm1.0}$ | $63.9_{\pm1.5}$ | $72.6_{\pm1.2}$ |
| | SC($\tau=0.6$) | $55.1_{\pm3.6}$ | $57.4_{\pm1.0}$ | $20.7_{\pm1.5}$ | $25.3_{\pm1.6}$ | $69.7_{\pm1.6}$ | $78.4_{\pm2.0}$ | $64.2_{\pm1.0}$ | $71.7_{\pm0.8}$ |
| | SC($\tau=0.8$) | $50.2_{\pm2.6}$ | $57.7_{\pm2.6}$ | $19.1_{\pm2.0}$ | $26.6_{\pm1.1}$ | $68.3_{\pm0.9}$ | $77.6_{\pm1.1}$ | $64.9_{\pm1.0}$ | $72.1_{\pm1.5}$ |
| | FIRE | $47.2_{\pm2.9}$ | $56.1_{\pm3.2}$ | $18.1_{\pm1.9}$ | $26.3_{\pm1.4}$ | $67.1_{\pm1.9}$ | $78.4_{\pm1.2}$ | $64.2_{\pm1.0}$ | $71.0_{\pm2.2}$ |
| | CoT-Decoding | $47.3_{\pm3.0}$ | $58.2_{\pm2.3}$ | $16.6_{\pm0.7}$ | $27.4_{\pm1.6}$ | $69.4_{\pm2.5}$ | $78.6_{\pm1.4}$ | $63.5_{\pm1.5}$ | $72.7_{\pm2.1}$ |
| | RAP | - | $58.6_{\pm1.8}$ | - | $27.6_{\pm1.2}$ | - | $79.4_{\pm1.1}$ | - | $72.4_{\pm1.3}$ |
| | *Soft Reasoning* | $\mathbf{61.4}_{\pm2.5}$ | $\mathbf{62.7}_{\pm1.0}$ | $\mathbf{25.8}_{\pm1.8}$ | $\mathbf{32.5}_{\pm1.5}$ | $\mathbf{72.2}_{\pm2.2}$ | $\mathbf{82.1}_{\pm1.2}$ | $\mathbf{66.1}_{\pm1.9}$ | $\mathbf{72.8}_{\pm1.5}$ |
| | **w/o** $r_{\text{verifier}}$ | $59.5_{\pm1.3}$ | $59.7_{\pm2.8}$ | $24.8_{\pm0.8}$ | $30.8_{\pm2.1}$ | $69.5_{\pm2.3}$ | $79.8_{\pm0.8}$ | $64.8_{\pm0.6}$ | $71.8_{\pm1.4}$ |
| | **w/o** $r_{\text{coherence}}$ | $61.2_{\pm2.3}$ | $60.3_{\pm2.5}$ | $25.5_{\pm3.3}$ | $29.5_{\pm2.3}$ | $70.2_{\pm1.6}$ | $80.0_{\pm1.0}$ | $65.5_{\pm1.7}$ | $72.0_{\pm1.3}$ |

**Setup & Hyperparameters.** Experiments are conducted in zero-shot and few-shot settings, with prompts including 1, 2, 4, and 8 exemplars for few-shot settings. To reduce variance, each configuration is repeated five times with different random seeds. We report the mean and standard deviation of accuracy across all runs. The convergence threshold is set to $0.01$.

### 4.2. Experimental Results

**Overall Performance.** Table 1 presents the accuracy of *Soft Reasoning* compared to baselines across four benchmarks and three LLMs under zero-shot and few-shot (8-shot) settings. The full table and the results for the Qwen2-70B-Ins model can be found in Tables 13 and 12, respectively, in Appendix B.8. Our approach consistently outperforms the best-performing baseline across different models, especially in the zero-shot setting (average improvement of 5% on GSM8K and 3% on GSM-Hard). Similar gains appear in the few-shot setting, where our method achieves the highest accuracy on most tasks and model variants. While effective, SC requires extensive hyperparameter tuning (e.g. varying temperature values) for each individual model and dataset to achieve optimal performance. In contrast, our more systematic search method improves solution quality consistently without the need for separate tuning in each scenario.

**Coverage Analysis.** For each method, we calculate the probability of covering the correct answer in at least one of the generated answers. Our approach consistently achieves the highest coverage across all models and datasets. For instance, on GSM8K with LLaMA3-8B-Ins in the zero-shot setting, our method attains 91.8% coverage, outperforming FIRE (84.5%) and CoT-Decoding (85.3%). Detailed coverage probabilities for all models and datasets can be found in Table 14 in Appendix B.8. These results demonstrate that our controlled exploration strategy effectively enhances the likelihood of generating correct answers, highlighting its robustness over traditional methods.

**Effect of Exploration with Embedding Perturbations and Bayesian Optimisation.** A natural question to consider is why adding noise to embeddings leads to more diverse answer generation than temperature tuning. We follow Naik et al. (2024) to investigate this from the perspective of neuron activations in the Transformer's MLP layers. As shown in Figure 3, applying our method increases the activation rate of neurons by roughly 3–4% in nearly all layers relative to the Self-Consistency (SC) baseline, suggesting that our perturbations stochastically trigger more diverse neural pathways.

To probe whether a specific subset of "critical neurons" may be responsible for correct reasoning, we identify neurons

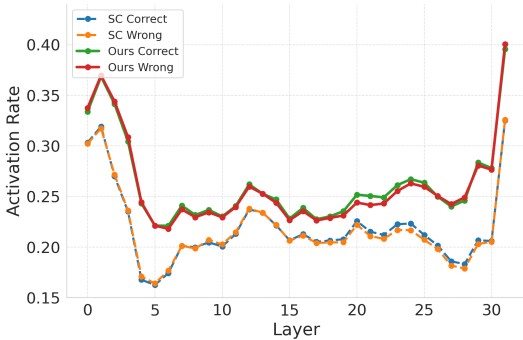

*Figure 3.* MLP layer activation rates across Transformer layers for the first five tokens of generated answers, sampled 200 times. The curves compare our method (*Soft Reasoning*) and the Self-Consistency (SC) baseline, further separated into correct and incorrect answers.

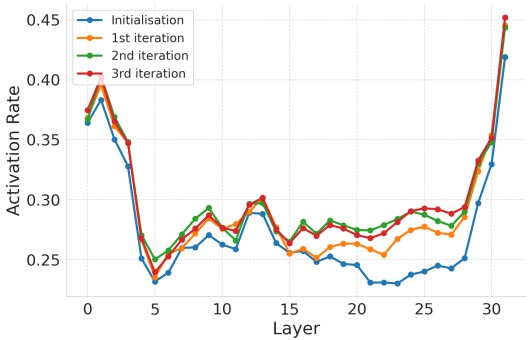

*Figure 4.* Activation rates of critical MLP neurons across Transformer layers during initialisation and iterations.

in a single sample whose activations exhibit the strongest correlation with correctness. In Figure 4, we track the activation rate of these critical neurons across our Bayesian optimisation iterations. We observe a steady increase, particularly in layers 15-30, suggesting that our iterative sampling and verification increasingly activates these key pathways.

Together, these findings support the hypothesis that our embedding perturbation and controlled exploration approach not only diversifies generation but also systematically uncovers and reinforces the neuron activations crucial for deriving correct answers. For more detailed experimental procedures, please refer to Appendix B.3.

**Convergence of Bayesian Optimisation.** Another question regarding our search algorithm is how quickly and reliably it converges. To investigate this, we track two key metrics across our Bayesian optimisation iterations: (1) The evolution of the correlation matrix among the sampled embedding points. As shown in Figure 5, the correlation matrix becomes more structured over iterations, showing higher correlations among top-performing candidates. (2) The pro-

portion of test examples that terminate after the $n^{\text{th}}$ iteration for each dataset in both zero-shot and few-shot settings, as reported in Table 2. With a maximum of 4 iterations, no search exceeds the fourth iteration, and only a small fraction require iteration 4. This rapid termination suggests that the EI-driven sampling strategy quickly identifies promising regions of the embedding space for most queries, minimising the need for further rounds of exploration.

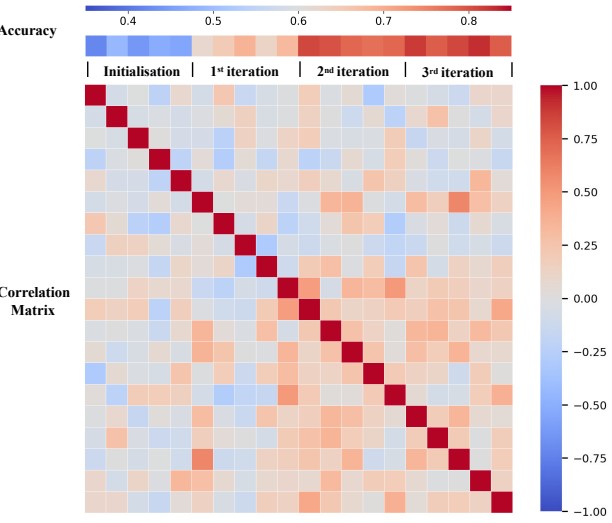

*Figure 5.* Visualisation of the correlation matrix evolution during Bayesian optimisation across iterations.

*Table 2.* Proportion (%) of test examples that terminate at the $n^{\text{th}}$ iteration for each dataset and setting using LLaMA.

| Shot | Iteration | GSM8K | GSM-Hard | SVAMP | StrategyQA |
|------|-----------|-------|----------|-------|-----------|
| Zero | 1 | $65.0_{\pm2.9}$ | $70.3_{\pm3.0}$ | $64.8_{\pm1.9}$ | $58.8_{\pm2.3}$ |
|      | 2 | $30.1_{\pm2.9}$ | $24.4_{\pm3.0}$ | $28.9_{\pm1.8}$ | $31.1_{\pm2.3}$ |
|      | 3 | $4.1_{\pm1.6}$ | $4.9_{\pm1.6}$ | $5.4_{\pm1.3}$ | $8.8_{\pm1.8}$ |
|      | 4 | $0.8_{\pm0.8}$ | $0.4_{\pm0.3}$ | $0.9_{\pm0.5}$ | $1.4_{\pm0.5}$ |
| Few  | 1 | $76.8_{\pm2.2}$ | $77.4_{\pm2.2}$ | $79.7_{\pm2.0}$ | $66.7_{\pm3.7}$ |
|      | 2 | $20.7_{\pm0.8}$ | $20.6_{\pm2.7}$ | $19.1_{\pm2.8}$ | $26.6_{\pm3.7}$ |
|      | 3 | $2.5_{\pm1.4}$ | $1.8_{\pm0.6}$ | $1.2_{\pm1.0}$ | $6.1_{\pm2.1}$ |
|      | 4 | $0.0_{\pm0.0}$ | $0.2_{\pm0.3}$ | $0.0_{\pm0.0}$ | $0.6_{\pm0.4}$ |

*Table 3.* Comparison of performance and token usage between our method and RAP across benchmarks.

| Category | Method | GSM8K | GSM-Hard | SVAMP | StrategyQA |
|----------|--------|-------|----------|-------|-----------|
| Result (%) | RAP | 80.7 | 32.7 | 87.9 | 73.4 |
|  | *Soft Reasoning* | **84.3** | **35.7** | **90.2** | **75.6** |
| Input Token Count | RAP | 25710.8k | 33152.1k | 15058.5k | 17426.2k |
|  | *Soft Reasoning* | 1457.1k | 1847.8k | 1172.8k | 1180.6k |
| Output Token Count | RAP | 334.1k | 402.5k | 241.2k | 274.1k |
|  | *Soft Reasoning* | 211.9k | 262.5k | 162.4k | 155.4k |
| Time (min) | RAP | 184.5 | 234.1 | 142.5 | 149.7 |
|  | *Soft Reasoning* | 23.2 | 28.4 | 18.4 | 17.4 |

**Efficiency and Performance Analysis.** Table 3 presents a comparison of our approach with RAP in both performance

and efficiency. Our method achieves better accuracy on all tasks while drastically reducing computational overhead. Specifically, our input token consumption averages only 6.19% of RAP's, our output token usage is 63.28% of RAP's, and our inference time is 14.3% of RAP's. These results highlight that our method not only improves accuracy but also substantially reduces token usage, thereby delivering superior overall efficiency. Additional analysis of inference time and memory usage is provided in Appendix B.5.

### 4.3. Ablation studies

**Objective Function.** To evaluate the importance of each reward component, we removed either the verifier score (**w/o** $r_{\text{verifier}}$) or the coherence term (**w/o** $r_{\text{coherence}}$) from our method. Table 1 compare these ablated variants with our full approach. In all tasks and model configurations, omitting either term degrades both accuracy and coverage, indicating that both components are vital. The verifier score clearly helps filter out incorrect or spurious solutions, while the coherence penalty ensures each output remains semantically consistent, particularly for complex or multi-step reasoning. Indeed, both correctness-guided verification and semantic coherence play essential roles in navigating the solution space effectively.

**Why Choose EI?** There are various acquisition functions for Bayesian Optimization (BO), such as the UCB score, Probability of Improvement (PI), and GP-UCB. While PI and GP-UCB are viable alternatives to Expected Improvement (EI), the cumulative regret bound for GP-UCB matches that of EI (Shahriari et al., 2015). In contrast, PI considers only the probability of improvement and ignores its magnitude, making it less theoretically grounded and more prone to premature exploitation (Srinivas et al., 2010).

*Table 4.* Performance comparison between different acquisition functions across datasets and settings using LLaMa.

| Shot | Method | GSM8K | GSM-Hard | SVAMP | StrategyQA |
|------|--------|-------|----------|-------|------------|
| **Zero** | EI (ours) | **79.4**±1.2 | **28.2**±1.8 | **88.2**±1.3 | **67.2**±0.7 |
| | PI | 74.6±1.5 | 28.0±1.5 | 85.3±1.0 | 66.9±1.8 |
| | UCB $\beta$=1 | 76.7±1.3 | 27.7±1.6 | 86.0±1.5 | 66.7±1.5 |
| | UCB $\beta$=2 | 77.9±1.9 | 27.8±0.8 | 85.0±1.0 | 66.8±2.3 |
| | UCB $\beta$=5 | 75.6±1.9 | 27.7±0.8 | 85.3±0.8 | 66.7±1.0 |
| **Few** | EI (ours) | **84.3**±1.4 | 35.7±1.0 | **90.2**±0.6 | 75.6±0.8 |
| | PI | 82.1±1.2 | 35.2±0.8 | 89.5±2.2 | 74.3±0.4 |
| | UCB $\beta$=1 | 83.3±0.3 | 34.7±1.2 | 88.0±0.0 | 74.7±1.5 |
| | UCB $\beta$=2 | 83.3±1.6 | **36.2**±1.5 | 88.7±1.4 | **75.7**±1.9 |
| | UCB $\beta$=5 | 81.8±0.9 | 34.2±2.5 | 89.7±0.8 | 75.0±0.7 |

As shown in Table 4, the experiments show that PI consistently underperforms compared to EI, as expected from the theoretical discussion above. For GP-UCB, its performance is sensitive to the choice of the exploration parameter and is, in most settings, worse than EI. We also note that the optimal parameter choice for GP-UCB varies across different

tasks, making it difficult to guarantee good performance in unseen settings. In contrast, EI performs robustly without requiring task-specific tuning.

**Optimisation Scope** To investigate how the number of optimised tokens affects performance, we conducted additional experiments where we optimised embeddings for the first $k$ tokens (instead of just the first token).

*Table 5.* Accuracy (%) when optimising embeddings for different numbers of initial tokens using LLaMa.

| Shot | #token ($k$) | GSM8K | GSM-Hard | SVAMP | StrategyQA |
|------|--------------|-------|----------|-------|------------|
| Zero | 1 (ours) | **79.4**±1.2 | **28.2**±1.8 | **88.2**±1.3 | **67.2**±0.7 |
| | 2 | 75.0±0.8 | 24.8±0.6 | 83.0±0.5 | 68.5±0.5 |
| | 5 | 69.7±3.5 | 22.2±1.0 | 85.5±0.3 | 66.3±0.6 |
| | 10 | 61.0±3.1 | 17.2±1.6 | 82.8±0.8 | 67.3±1.9 |
| | 20 | 52.2±2.3 | 19.0±1.3 | 74.3±0.6 | 67.8±2.5 |
| Few | 1 (ours) | **84.3**±1.4 | **35.7**±1.0 | **90.2**±0.6 | **75.6**±0.8 |
| | 2 | 83.3±1.3 | 34.7±1.0 | 90.2±1.4 | 74.2±1.3 |
| | 5 | 81.2±2.1 | 29.8±0.8 | 88.3±0.3 | 71.7±1.2 |
| | 10 | 73.7±2.3 | 23.8±0.3 | 86.8±0.8 | 71.7±1.6 |
| | 20 | 62.0±2.6 | 18.2±1.9 | 81.5±2.0 | 68.7±1.1 |

As shown in Tab 5, the performance generally degrades as $k$ increases, especially beyond 5 tokens. This suggests that naively extending to multiple tokens can introduce instability or overfitting. We also compared with RAP (one of our baselines), a tree-search-based method that operates at the sequence level rather than token-by-token—though it shares similar ideas with token-wise search. While RAP achieves strong performance, it incurs substantially higher cost and still underperforms our approach. Developing a multi-token optimisation strategy that can achieve both high accuracy and cost-effectiveness would require deeper investigation and extensive experimentation.

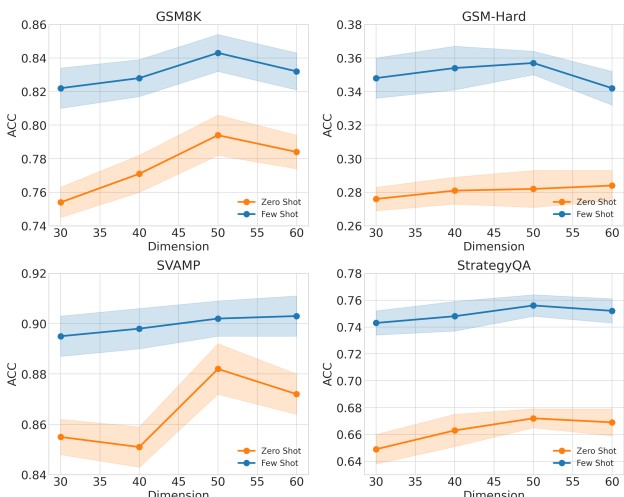

*Figure 6.* Accuracy (solid lines) and standard deviation (shaded areas) across reduced dimensions.

**Impact of Lower-Dimensional Space Dimensionality.**
We evaluate our Bayesian optimisation approach under various reduced dimensions before mapping back to the full embedding space (Figure 6). Across all four tasks and both zero- and few-shot settings, performance tends to improve up to $d = 50$. Although increasing $d$ to 60 sometimes yields a small additional gain, the differences are minor, and $d = 50$ consistently achieves near-best or best results. To further evaluate the robustness of random projection, we conducted additional stability experiments, which are presented in Appendix B.4.

**Impact of Special Token Placement.** We compare three ways of inserting the perturbed special token into the prompt: at the beginning (*First*), somewhere in the middle (*Middle*), or as an appended token (*Last*). Table 6 shows for both zero-shot and few-shot settings, placing the special token at the end of the prompt (*Last*) generally yields higher accuracy and better coverage. One possible explanation is that placing the special token last ensures minimal disruption to the original semantics of the prompt, while still allowing *Soft Reasoning* to alter the initial token embedding and induce sufficiently diverse generation pathways. Based on this observation, we adopt the *Last* placement strategy in all subsequent experiments.

*Table 6.* Comparison of accuracy and coverage across different special token placements using LLaMa.

| Type | Shot | Iteration | GSM8K | GSM-Hard | SVAMP | StrategyQA |
|---|---|---|---|---|---|---|
| **Result** | Zero | First | $77.7_{\pm2.5}$ | $25.5_{\pm1.3}$ | $85.0_{\pm0.9}$ | $67.0_{\pm1.3}$ |
| | | Middle | $78.5_{\pm3.1}$ | $27.3_{\pm1.3}$ | $84.0_{\pm0.9}$ | $\mathbf{67.8}_{\pm4.4}$ |
| | | Last (ours) | $\mathbf{79.4}_{\pm1.2}$ | $\mathbf{28.2}_{\pm1.8}$ | $\mathbf{88.2}_{\pm1.3}$ | $67.2_{\pm0.7}$ |
| | Few | First | $82.1_{\pm0.8}$ | $29.0_{\pm3.5}$ | $89.2_{\pm0.3}$ | $74.1_{\pm0.8}$ |
| | | Middle | $82.3_{\pm0.8}$ | $32.3_{\pm1.8}$ | $89.7_{\pm1.3}$ | $74.0_{\pm3.0}$ |
| | | Last (ours) | $\mathbf{84.3}_{\pm1.4}$ | $\mathbf{35.7}_{\pm1.0}$ | $\mathbf{90.2}_{\pm0.6}$ | $\mathbf{75.6}_{\pm0.8}$ |
| **Coverage** | Zero | First | $85.8_{\pm2.3}$ | $31.2_{\pm2.8}$ | $93.0_{\pm1.8}$ | $92.7_{\pm0.0}$ |
| | | Middle | $89.5_{\pm2.6}$ | $32.7_{\pm0.8}$ | $93.3_{\pm1.0}$ | $93.1_{\pm0.3}$ |
| | | Last (ours) | $\mathbf{91.8}_{\pm1.4}$ | $\mathbf{37.0}_{\pm1.5}$ | $93.8_{\pm0.4}$ | $\mathbf{93.7}_{\pm1.3}$ |
| | Few | First | $92.0_{\pm0.5}$ | $40.8_{\pm1.2}$ | $94.7_{\pm1.2}$ | $93.4_{\pm0.9}$ |
| | | Middle | $92.2_{\pm1.0}$ | $44.7_{\pm2.5}$ | $94.5_{\pm0.5}$ | $93.1_{\pm0.8}$ |
| | | Last (ours) | $\mathbf{92.2}_{\pm0.8}$ | $\mathbf{49.8}_{\pm1.0}$ | $\mathbf{95.8}_{\pm1.2}$ | $93.3_{\pm1.8}$ |

**Verifier Comparison: Judgement vs. Generation.** Inspired by recent work suggesting that LLMs can be more adept at *generating* correct outputs than critiquing existing ones (Miao et al., 2024; Zhang et al., 2024), we explore four verifier strategies: *Single-Judge*, which evaluates each candidate independently; *Single-Generate*, which regenerates a purportedly correct answer for each candidate; *Multi-Judge*, which scores multiple candidates collectively; and *Multi-Generate*, which produces a new solution from multiple candidates, labeling any matching candidate as correct. Given its consistently strong performance across settings, we select *Multi-Generate* as the default verifier in our experiments. The detailed definitions of these prompt templates are provided in Appendix B.6.

*Table 7.* Binary classification accuracy (%) of different verifier strategies, each determining whether a generated answer is correct. *Single* strategies judge or generate in isolation per answer, while *Multi* strategies consider multiple candidate solutions together.

| Verifier | GSM8K | GSM-Hard | SVAMP | StrategyQA |
|---|---|---|---|---|
| Single-Judge | 75.9 | 60.9 | 82.7 | 63.9 |
| Multi-Judge | 80.4 | 46.8 | 87.5 | 67.3 |
| Single-Generate | 78.0 | 40.7 | 82.7 | 71.7 |
| Multi-Generate (ours) | **87.6** | **78.2** | **93.4** | **78.9** |

Table 7 reports the binary classification accuracies for each verifier. *Multi-Generate* yields the highest verification accuracy on all datasets. This indicates that leveraging the model's generative capabilities leads to more reliable correctness assessment.

*Table 8.* Final accuracy (%) achieved by using different verifier strategies in our overall framework using LLaMa.

| Shot | Verifier | GSM8K | GSM-Hard | SVAMP | StrategyQA |
|---|---|---|---|---|---|
| Zero | Single-Judge | $76.3_{\pm1.5}$ | $28.1_{\pm1.6}$ | $83.0_{\pm1.4}$ | $63.0_{\pm1.7}$ |
| | Multi-Judge | $77.4_{\pm2.2}$ | $26.5_{\pm2.2}$ | $86.5_{\pm1.3}$ | $67.1_{\pm0.8}$ |
| | Single-Generate | $76.5_{\pm1.1}$ | $27.6_{\pm2.1}$ | $84.3_{\pm0.0}$ | $66.4_{\pm0.0}$ |
| | Multi-Generate | $\mathbf{79.4}_{\pm1.2}$ | $\mathbf{28.2}_{\pm1.8}$ | $\mathbf{88.2}_{\pm1.3}$ | $\mathbf{67.2}_{\pm0.7}$ |
| Few | Single-Judge | $82.4_{\pm1.5}$ | $35.0_{\pm1.4}$ | $89.6_{\pm1.4}$ | $72.0_{\pm1.3}$ |
| | Multi-Judge | $82.5_{\pm1.3}$ | $\mathbf{36.1}_{\pm2.1}$ | $90.1_{\pm0.9}$ | $73.2_{\pm1.2}$ |
| | Single-Generate | $82.4_{\pm1.3}$ | $34.5_{\pm1.4}$ | $89.7_{\pm1.2}$ | $74.7_{\pm1.2}$ |
| | Multi-Generate | $\mathbf{84.3}_{\pm1.4}$ | $35.7_{\pm1.0}$ | $\mathbf{90.2}_{\pm0.6}$ | $\mathbf{75.6}_{\pm0.8}$ |

We compare final solution accuracy in Table 8. While single verifier judge or generate answers individually, multi-candidate generation leads to the highest end-to-end performance. *Multi-Generate* outperforms alternatives in both zero-shot and few-shot settings, harnessing the model's generative capacity more effectively than judgment-based verifiers. Notably, even substituting simpler verifiers keeps our framework competitive with strong baselines, underscoring the robustness and efficacy of generation-based verification. Details on how the number of sampled embeddings $k$ influences performance are provided in Appendix B.7.

## 5. Conclusions and Future Directions

We introduce an embedding-based optimisation framework that enhances LLM reasoning by refining the first-token embedding. By integrating controlled perturbations with Bayesian optimisation, *Soft Reasoning* improves accuracy, is model-agnostic, and remains computationally efficient. Our approach relies on a verifier that may provide unreliable feedback, impacting optimisation. It also operates at the token level, which poses challenges for interpreting how perturbations influence reasoning. Future work will focus on improving verifier reliability, extending optimisation beyond the first token, and enhancing interpretability to better understand perturbation effects on reasoning.

## Impact Statement

This paper presents work whose goal is to advance the field of Machine Learning. There are many potential societal consequences of our work, none of which we feel must be specifically highlighted here.

## Acknowledgments

This work was supported in part by the UK Engineering and Physical Sciences Research Council (EPSRC) through a Turing AI Fellowship (grant no. EP/V020579/1, EP/V020579/2) and a New Horizons grant (grant no. EP/X019063/1), and KCL's Impact Acceleration Account (grant no. EP/X525571/1). A PhD studentship from the Chinese Scholarship Council funds Qinglin Zhu. The authors also acknowledge the use of the King's Computational Research, Engineering, and Technology Environment (CREATE) at King's College London. We would also like to thank Dr. Han Zhang and Professor Tao Gui for their valuable suggestions during the rebuttal period.

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

## A. Methodology Supplement

### A.1. Technical Details on Optimisation Objective

Although we simply sum the two components, they naturally operate in a progressive, tie-breaking manner. If two candidate outputs differ in their verifier scores, the one with the higher $r_{\text{verifier}}$ value immediately results in a higher $f(x)$. Only when the verifier scores are identical does $r_{\text{coherence}}$ serve to break the tie, making the process effectively hierarchical, even in this additive form.

### A.2. Technical Details on Bayesian Optimisation

In constructing the prior distribution, for a finite collection of points $x_{1:n}$, the prior joint distribution on them is:

$$f(x_{1:n}) \sim \mathcal{N}(\mu_0(x_{1:n}), \Sigma_0(x_{1:n}, x_{1:n})),$$

where $f(x_{1:n}) = [f(x_1), \ldots, f(x_n)]^\top$, $\mu_0(x_{1:n}) = [\mu_0(x_1), \ldots, \mu_0(x_n)]^\top$, and the covariance matrix is:

$$\Sigma_0(x_{1:n}, x_{1:n}) = \begin{bmatrix} \Sigma_0(x_1, x_1) & \ldots & \Sigma_0(x_1, x_n) \\ \vdots & \ddots & \vdots \\ \Sigma_0(x_n, x_1) & \ldots & \Sigma_0(x_n, x_n) \end{bmatrix}.$$

We set $\mu_0(x) = 0$, indicating no additional preference for function values at the prior stage. For the covariance matrix $\Sigma_0(x_i, x_j)$, we use the Gaussian kernel with bandwidth $\ell$:

$$\Sigma_0(x_i, x_j) = k(x_i, x_j) = \exp\left(-\frac{\|x_i - x_j\|^2}{2\ell^2}\right).$$

The kernel is chosen such that when $x_i$ and $x_j$ are close, the kernel value $k(x_i, x_j)$ approaches 1, indicating strong correlation, and when they are far apart, the kernel value approaches 0, reflecting weak correlation.

After observing $f(x_{1:k})$, we aim to infer the value of $f(x)$ at a new point $x$. Using Bayes' rule (Rasmussen & Williams, 2006), we update the posterior distribution of $f(x)$ conditioned on these observed values:

$$f(x) \mid f(x_{1:k}) \sim \mathcal{N}(\mu_k(x), \sigma_k^2(x)),$$

where the posterior mean and variance are given by

$$\mu_k(x) = \Sigma_0(x, x_{1:k})\Sigma_0(x_{1:k}, x_{1:k})^{-1}(f(x_{1:k}) - \mu_0(x_{1:k})) + \mu_0(x),$$

$$\sigma_k^2(x) = \Sigma_0(x, x) - \left[\Sigma_0(x, x_{1:k})\Sigma_0(x_{1:k}, x_{1:k})^{-1} \Sigma_0(x_{1:k}, x)\right].$$

### A.3. Maximising Expected Improvement

To select the point $x$ that maximises the expected improvement $\text{EI}_k(x)$, we adopt a sampling-based approach. Specifically, we randomly sample a large number of candidate

points from a standard normal distribution. In our experiments, we generate 5000 points $x_1, x_2, \ldots, x_{5000}$, where each point $x_i \in \mathbb{R}^D$ is sampled as:

$$x_i \sim \mathcal{N}(0,1)^D, \quad i = 1, 2, \ldots, 5000.$$

For each sampled point, we compute the expected improvement $\text{EI}_k(x)$ using (3). After evaluating the expected improvement for all sampled points, we select the point with the maximum $\text{EI}_k(x)$ as the next candidate for evaluation. This method provides an efficient and practical way to approximate the global maximum of $\text{EI}_k(x)$ within the sampling budget.

### A.4. Adaptive Expected Improvement

In defining $f(x)$, we assume an ideal verifier with perfect accuracy, meaning it provides an error-free assessment of correctness. However, in practice, the verifier's accuracy is less than 1, introducing uncertainty into its evaluations. This results in noisy observations, where the observed score $o_k$ deviates from the true function value $f(x_k)$. We model this noise as:

$$o_k = f(x_k) + \eta_k, \quad \eta_k \sim \mathcal{N}(0, \lambda I),$$

where $\lambda$ is an objective noise constant, determined by the inherent noise level. To address this noise in Bayesian optimisation, we use an adaptive version of the EI acquisition function that explicitly accounts for the uncertainty in observations:

$$\text{EI}_k(x) = \left[\mu_k(x) - f_k^*\right]^+ + \omega_k \sigma_k(x) \phi\left(\frac{\mu_k(x) - f_k^*}{\omega_k \sigma_k(x)}\right)$$
$$- \left|\mu_k(x) - f_k^*\right| \Phi\left(\frac{\mu_k(x) - f_k^*}{\omega_k \sigma_k(x)}\right),$$

where $\omega_k = \sqrt{\gamma_k + 1 + \ln(1/\delta)}$ is the noise-adaptive scaling factor, and information gain term $\gamma_k$ is defined as:

$$\gamma_k = \max I(o(x_{1:k}); f(x_{i:k}))$$
$$= \frac{1}{2} \log \det(I + \lambda^{-1} \Sigma_0(x_{1:k}, x_{1:k})).$$

$I(o(x_{1:k}); f(x_{i:k}))$ represents the mutual information between the function values and the noisy observations, and $\delta$ is a hyperparameter in $(0, 1)$, controlling the balance between exploration and exploitation. This adaptation is inspired by the following observation (Vakili et al., 2021; Tran-The et al., 2022):

**Theorem A.1.** *For any choice of $\delta \in (0, 1)$, with probability at least $1 - \delta$, the cumulative regret $R_T$ satisfies*

$$R_T := \sum_{k=1}^{T} \left[f_k^* - f(x_k)\right] = O\left(\gamma_T \sqrt{T}\right).$$

This sublinear bound ensures that, with high probability, the regret grows at a slower rate than the number of iterations, thereby guaranteeing the convergence of the optimisation process.

We set $\delta = 0.1$, ensuring a 90% probability of convergence while balancing exploration and exploitation. A smaller $\delta$ (e.g., 0.01) strengthens theoretical guarantees but increases exploration, slowing convergence. A larger $\delta$ (e.g., 0.2) favours exploitation, accelerating convergence but weakening guarantees. Our choice provides stability and efficiency without excessive exploration.

### A.5. Addressing the Curse of Dimensionality in Bayesian Optimisation

Traditional Bayesian optimisation struggles in high-dimensional spaces due to the curse of dimensionality, making it impractical for embedding vectors used in LLMs. To address this challenge, we leverage the following result from high-dimensional optimisation (Wang et al., 2016; Nayebi et al., 2019), which allows us to perform optimisation in a lower-dimensional space and to subsequently map points back to the original space.

**Theorem A.2.** *Let $D$ be the dimension of the embedding vectors. A function $f : \mathbb{R}^D \to \mathbb{R}$ is defined to have an effective dimensionality $d_e$, with $d_e \leq D$, if the following condition is satisfied: $\exists$ a subspace $E$ of dimension $d_e$, such that $\forall x_E \in E \subset \mathbb{R}^D$ and $x_\perp \in E^\perp \subset \mathbb{R}^D$, where $E^\perp$ is the orthogonal complement of $E$, the function satisfies: $f(x_E + x_\perp) = f(x_E)$. In other words, $d_e$ is the smallest dimension that retains all variability of $f$. Now, for $d \geq d_e$, consider a random matrix $A \in \mathbb{R}^{D \times d}$ with independent $\mathcal{N}(0,1)$ entries. Then,*

$$\forall x \in \mathbb{R}^D, \exists u \in \mathbb{R}^d \text{ such that } f(x) = f(Au).$$

This result implies that for any optimiser $x^* \in \mathbb{R}^D$, there exists a corresponding point $u^* \in \mathbb{R}^d$ such that $f(x^*) = f(Au^*)$. Therefore, instead of performing optimisation in the high-dimensional space, we can optimise the function $g(u) := f(Au)$ in the lower-dimensional space.

Suppose our initial sampled points and their evaluations are denoted by the set $\mathcal{U}_k = \{(u_1, g(u_1)), \ldots, (u_k, g(u_k))\}$, where each $u_i \in \mathbb{R}^d$ has independent standard normal entries. Similarly, we initialise $A \in \mathbb{R}^{D \times d}$ as a random matrix with independent $\mathcal{N}(0, 1)$ entries. We then find the next point to sample $u_{k+1} \in \mathbb{R}^d$ by optimising the acquisition function:

$$u_{k+1} = \operatorname*{argmax}_{u \in \mathbb{R}^d} EI_k(u \mid \mathcal{U}_k),$$

where $EI_k(\cdot \mid \mathcal{U}_k)$ represents the Expected Improvement conditioned on the current dataset $\mathcal{U}_k$ with the objective

function being $g(\cdot)$ on $\mathbb{R}^d$. Then, we augment the dataset

$$\mathcal{U}_{k+1} := \mathcal{U}_k \cup \{(u_{k+1}, f(Au_{k+1}))\},$$

and iterate.

### A.6. Setting the Convergence Threshold

The convergence threshold $\epsilon$ determines when the algorithm stops iterating. While smaller thresholds generally lead to more precise results, they can also increase computational costs due to additional iterations. We set $\epsilon = 0.01$ in our experiments, as this value strikes a balance between convergence quality and computational efficiency.

## B. More Experimental Details

### B.1. Experiment Settings

We evaluate all methods on four benchmark datasets (GSM8K, GSM-Hard, SVAMP, and StrategyQA) using 200 randomly sampled test examples per dataset, with a maximum output length of 300 tokens. For SC, FIRE, and CoT-Decoding, we follow baseline settings with a temperature of 0.8 and sample 5 outputs per example for majority voting. For RAP, we adopt the original paper's settings (Hao et al., 2023), using Monte Carlo Tree Search with 4 actions, a confidence threshold of 8, a depth limit of 5, and 10 search iterations.

### B.2. Comparison with Controlled Generation Methods

We additionally compare our method against two recent controlled generation approaches: Trainable Prefix Scorers (Mudgal et al., 2024) and Constrained fine-tuning (Qi et al., 2025a).

The prefix scorer method uses trainable scorers to guide decoding, while constrained fine-tuning proposes a fine-tuning objective aimed at improving robustness against adversarial prompts. Both require additional training, making direct comparison with our training-free method less straightforward. For a fair comparison, we implemented baselines without extra training where possible.

| Method | Training | Shot | GSM8K | GSM-Hard | SVAMP | StrategyQA |
|---|---|---|---|---|---|---|
| Constrained Fine-tuning | ✓(LoRA) | - | $78.3_{\pm0.7}$ | $13.6_{\pm0.5}$ | $83.5_{\pm0.7}$ | $\mathbf{81.3}_{\pm0.8}$ |
| Prefix Scorer | ✗ | Zero | $75.2_{\pm0.9}$ | $26.1_{\pm1.3}$ | $83.6_{\pm0.9}$ | $65.2_{\pm1.3}$ |
| *Soft Reasoning* | ✗ | Zero | $\mathbf{79.4}_{\pm1.2}$ | $\mathbf{28.2}_{\pm1.8}$ | $\mathbf{88.2}_{\pm1.3}$ | $67.2_{\pm0.7}$ |
| Prefix Scorer | ✗ | Few | $81.2_{\pm1.6}$ | $33.6_{\pm1.4}$ | $88.5_{\pm1.2}$ | $72.4_{\pm1.1}$ |
| *Soft Reasoning* | ✗ | Few | $\mathbf{84.3}_{\pm1.4}$ | $\mathbf{35.7}_{\pm1.0}$ | $\mathbf{90.2}_{\pm0.6}$ | $75.6_{\pm0.8}$ |

*Table 9.* Accuracy (%) comparison with trainable prefix scorers and constrained fine-tuning methods. Our method requires no additional training.

As shown, while constrained fine-tuning achieves the best performance on StrategyQA with additional training, our method—without requiring any extra training—achieves

superior results on GSM8K, GSM-Hard, and SVAMP, particularly in the few-shot setting. In a fairer comparison (without any training), our method consistently outperforms the prefix scorer across all tasks.

### B.3. Additional Details on Neuron Activation Analysis

#### B.3.1. OVERALL ACTIVATION RATE

We analyse the neuron activations in the GLU-based MLP layers of the LLaMA model (Naik et al., 2024). Specifically, the hidden representation in the $i$-th layer is computed as:

$$h^i = \left(\mathrm{act\_fn}(\tilde{h}^i W_1^i) \otimes \tilde{h}^i W_3^i\right) \cdot W_2^i, \qquad (4)$$

where $\otimes$ denotes element-wise multiplication, and $\mathrm{act\_fn}(\cdot)$ is a non-linear activation function. We consider the $j$-th neuron inside the $i$-th FFN layer *activated* if its activation value $\left[\mathrm{act\_fn}(\tilde{h}^i W_1^i)\right]_j$ exceeds zero.

We compare Self-Consistency (SC) (Wang et al., 2023c) with our Bayesian-optimisation-based perturbation method. For a single question (i.e., same prompt), we generate 200 samples using LLaMA and record neuron activations for each of the first 5 output tokens. We then visualise and compare the average activation rates between SC and our approach in Figure 3.

#### B.3.2. KEY NEURON IDENTIFICATION AND VERIFICATION

We identify key neurons by analysing activation rates in SC-generated samples. We first separate these samples into correct and incorrect categories. For each neuron $j$ in layer $i$, we calculate the activation difference:

$$\Delta_{i,j} = \mathrm{avg}_{\mathrm{correct}}(a_{i,j}) - \mathrm{avg}_{\mathrm{incorrect}}(a_{i,j}),$$

where $a_{i,j}$ denotes the activation value of the $j$-th neuron in the $i$-th layer. We first select the top 25% of neurons with the highest activation frequency, then rank these by $\Delta_{i,j}$, and select the top 20% with the largest positive values, resulting in 5% of all neurons as "key neurons."

This statistical approach to identifying critical neurons is grounded in prior research demonstrating that it is possible to trace information flow within transformers and isolate neurons with causal influence on model predictions. Such studies have used targeted interventions like activation replacement or ablation to validate neuron importance (Dai et al., 2022; Meng et al., 2022).

To validate the functional significance of the identified key neurons, we conducted targeted masking experiments following this line of work. Masking the critical neurons identified for each input led to a significant drop in accuracy from 62.14% to 13.27%. As a control, we randomly masked an equivalent number of neurons under identical settings, resulting in a considerably higher average accuracy of 41.89%.

The substantial gap (41.89% → 13.27%) confirms that the identified neurons play an essential role in the model's reasoning process, beyond what would be expected by chance.

### B.4. Random Projection Stability

While dimensionality reduction alleviates the curse of dimensionality, it may lead to information loss. To examine whether the negative impacts of dimensionality reduction via random projection are controllable, we conducted additional validation experiments.

We tested the stability of random projection by running simulations using 50 different random projection matrices. For each run, we applied Bayesian optimisation under the same settings and recorded the final performance. The experiments were conducted on the GSM8K dataset using the LLaMA3-8B-Ins model. The results are summarised in Figure 7.

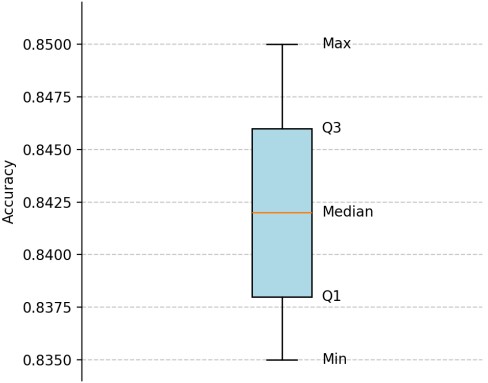

*Figure 7.* Distribution of final accuracy across 50 random projection matrices. Performance remains stable, indicating that random projection does not introduce significant variance.

The results indicate that performance remains stable across different random projections, with minimal variance between runs. This demonstrates that although some degree of information loss is inevitable, it does not introduce significant instability into the optimisation process. The chosen dimension ($d = 50$) offers a good balance between performance and computational efficiency.

### B.5. Computational Efficiency

We supplement the token count statistics with comparisons between our method and RAP on inference time and memory usage. To evaluate the latter, we focus on the two variable components: (1) KV cache, and (2) intermediate activations, since model weights remain constant across methods. Using vLLM's block-based memory tracking, we report both average and peak usage, sampled at 1-second intervals.

The results show that our method's inference time is only

| Dataset | Method | Time (min) | Intermediate Act. (avg, MB) | Intermediate Act. (peak, MB) | KV Cache (avg, MB) | KV Cache (peak, MB) |
|---|---|---|---|---|---|---|
| GSM8K | RAP | 184.52 | 1628.7 | 1874.4 | 252.5 | 568.0 |
| | *Soft Reasoning* | **23.15** | **1137.2** | **1178.1** | **176.5** | **312.0** |
| GSM-Hard | RAP | 234.14 | 1985.4 | 2354.9 | 426.5 | 574.0 |
| | *Soft Reasoning* | **28.42** | **881.2** | **1096.2** | **254.6** | **336.0** |
| SVAMP | RAP | 142.52 | 1464.8 | 2089.5 | 384.5 | 494.0 |
| | *Soft Reasoning* | **18.41** | **932.4** | **1393.2** | **185.8** | **296.0** |
| StrategyQA | RAP | 149.73 | 1833.5 | 1935.9 | 241.4 | 376.0 |
| | *Soft Reasoning* | **17.44** | **748.0** | **932.4** | **118.0** | **264.0** |

*Table 10.* Comparison of inference time and memory usage between RAP and our method.

12.30% of RAP's while also consuming significantly less memory, further validating its computational efficiency advantage.

We also report inference time (minutes) across all baselines:

| Method | GSM8K | GSM-Hard | SVAMP | StrategyQA |
|---|---|---|---|---|
| SC($\tau$=0.4) | 26.58 | 33.91 | 20.54 | 20.04 |
| SC($\tau$=0.6) | 26.12 | 34.46 | 21.76 | 19.87 |
| SC($\tau$=0.8) | 27.28 | 34.86 | 21.16 | 20.80 |
| FIRE | 26.70 | 32.26 | 21.59 | 20.17 |
| CoT-Decoding | 26.56 | 32.55 | 21.53 | 20.60 |
| RAP | 184.52 | 234.14 | 142.52 | 149.73 |
| *Soft Reasoning* | **23.15** | **28.42** | **18.41** | **17.44** |

*Table 11.* Inference time (minutes) comparison across baselines.

Our method consistently achieves the lowest inference time across all tasks, further demonstrating its efficiency beyond token-level savings.

### B.6. Prompts for Verifiers

We provide four prompt templates corresponding to the verifier strategies introduced in Section 4.3. These templates illustrate how each verifier approach is instantiated:

**Single-Judge.** This prompt asks the model to evaluate the correctness of a single final answer. The user provides a question, along with a final answer, and the verifier must decide whether that answer is correct.

**Multi-Judge.** This prompt provides multiple candidate answers and asks the verifier to assess their correctness collectively. The user includes each candidate's reasoning, and the verifier classifies which answers are correct.

**Single-Generate.** The prompt demonstrates a scenario where the verifier itself is prompted to *re-generate* the correct solution for one candidate. If the newly generated solution matches the candidate's answer, the candidate is deemed correct.

**Multi-Generate (Ours).** This prompt processes all candidate answers together and generates a new solution it believes to be correct. Any candidate matching this newly

generated solution is labeled as correct. This leverages the model's generative capacity more thoroughly than pure classification, yielding better verification accuracy in practice.

## B.7. Effect of the Number of Samples $k$.

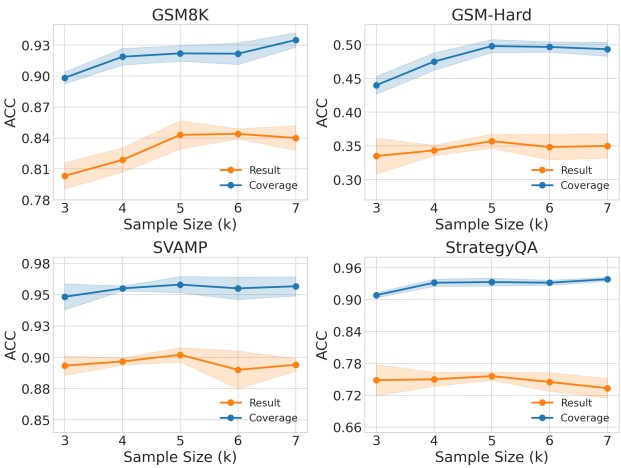

*Figure 8.* Accuracy (solid lines) and standard deviation (shaded areas) across different sample sizes $k$.

We investigate how the number of sampled embeddings $k$ in each iteration influences final performance. Figure 8 shows that as $k$ increases from 3 to 5, the final accuracy and coverage steadily rise for all evaluated tasks, although gains tend to plateau or fluctuate slightly after $k = 3$. This upward trend suggests that a moderate increase in $k$ promotes better exploration of potentially correct solutions in the embedding space. Based on these observations, we adopt $k = 5$ as our default setting, balancing solution diversity with computational cost.

## B.8. Additional Results

Tables 13 and 14 present the full experimental results for all methods and configurations across the four benchmarks and three LLMs (LLaMA3-8B-Instruct, Qwen2-7B-Instruct, and Mistral-7B-Instruct). These tables extend the summary reported in the main text, showing detailed accuracy, coverage, and standard deviations for zero-shot to 8-shot setups. They provide a comprehensive view of how each baseline and our approach perform under various hyperparameter and prompt configurations.

**Prompt for Single-Judge**

Based on the given question and the previous answers, please provide your judgment on the correctness of the final answer.

Question:
Jessie currently weighs 9 kilograms. After she started to go jogging every day, she lost 62 kilograms in the first week and 140 kilograms in the second week. How much did she weigh before starting to jog?

Answer:
$9 + 62 + 140 = 211$. So, Jessie weighed 211 kilograms after 2 weeks of jogging. Since she weighed 9 kilograms initially, she weighed $211 - 9 = 202$ kilograms before starting to jog. Answer: 202

Correct:
0

**Question:**
{User Question}

**Answer:**
{Previous answers}

**Correct:**

---

**Prompt for Multi-Judge**

Based on the given question and the previous answers, please provide your judgment on the correctness of the final answer.

Question:
Jack is stranded on a desert island. He wants some salt to season his fish. He collects 2 liters of seawater in an old bucket. If the water is 20% salt, how many ml of salt will Jack get when all the water evaporates?

Your previous answers:
0. Thought: 1250 ml of water evaporates, leaving 1000 ml of salt. Answer: 1000
1. Thought: The total amount of water is 2 liters = 2000 ml. The amount of salt is 20% of 2000 ml = $0.20 \times 2000$ ml = $\langle\langle 0.20 \times 2000 = 400\rangle\rangle 400$ ml. Answer: 400
2. Thought: 20% of 2 liters is $2 \times \frac{20}{100} = \langle\langle 2 \times 20/100 = 0.4\rangle\rangle 0.4$ liters. Since there are 1000 ml in 1 liter, 0.4 liters is $0.4 \times 1000 = \langle\langle 0.4 \times 1000 = 400\rangle\rangle 400$ ml. Answer: 400
3. Thought: 1 liter of seawater is 20% salt. So, 1 liter of seawater has $20\% \times 1$ liter = $\langle\langle 20 \times 0.1 = 0.2\rangle\rangle 0.2$ liters of salt. Since Jack has 2 liters of seawater, he will get $0.2 \times 2 = \langle\langle 0.2 \times 2 = 0.4\rangle\rangle 0.4$ liters of salt. Since there are 1000 ml in 1 liter, Jack will get $0.4 \times 1000 = \langle\langle 0.4 \times 1000 = 400\rangle\rangle 400$ ml of salt. Answer: 400
4. Thought: 20% of 2 liters is $2 \times \frac{20}{100} = \langle\langle 2 \times 20/100 = 0.4\rangle\rangle 0.4$ liters. There are 1000 ml in 1 liter, so 0.4 liters is $0.4 \times 1000 = \langle\langle 0.4 \times 1000 = 400\rangle\rangle 400$ ml. Answer: 400

Correct:
1, 2, 3, 4

**Question:**
{User Question}

**Your previous answers:**
{Previous answers}

**Correct:**

---

**Prompt for Single-Generate**

Based on the given question and the previous answers, please provide your analysis and final answer, starting the final answer with "Answer:"

Question:
Jack is stranded on a desert island. He wants some salt to season his fish. He collects 2 liters of seawater in an old bucket. If the water is 20% salt, how many ml of salt will Jack get when all the water evaporates?

Your previous answers:
1250 ml of water evaporates, leaving 1000 ml of salt. Answer: 1000

Analysis:
Let's think step by step. Jack has 2 liters of seawater, and 20% of it is salt. 2 liters = 2000 ml, so the amount of salt is 20% of 2000 ml = $0.20 \times 2000 = 400$ ml of salt.

Answer:
400

**Question:**
{User Question}

**Your previous answers:**
{Previous answers}

**Analysis:**

---

---

**Prompt for Multi-Generate (ours)**

Based on the given question and the previous answers, please provide your analysis and final answer, starting the final answer with "Answer:"

Question:
Artemis is making tea for a party. She knows her mom drinks an 8-ounce cup of tea and uses one ounce of tea. She will use this same ratio for the party. The party has 12 people there and each of them wants a 6-ounce cup of tea. How many ounces of tea does she need?

Your previous answers:
0. Thought: 8 ounces of tea for 1 cup, so 1 ounce of tea for $\frac{1}{8}$ of a cup. For 12 people, she needs $12 \times \frac{6}{8} = 9$ ounces of tea. Answer: 9
1. Thought: 6 ounces of tea is needed for each person. Since there are 12 people, $12 \times 6 = 72$ ounces of tea are needed. Since each ounce of tea is used for 1 cup, 72 ounces of tea will make 72 cups of tea. Answer: 72
2. Thought: 6 ounces of tea is $\frac{6}{8} = \frac{3}{4}$ of an 8-ounce cup. For 12 people, she needs $12 \times \frac{3}{4} = 9$ ounces of tea. Answer: 9
3. Thought: $12 \times 6 = 72$ ounces of tea needed. Since each ounce of tea is used for 1 cup, Artemis needs 72 ounces of tea. Answer: 72
4. Thought: 8 ounces of tea is used for 1 cup. So for 6 ounces of tea, she will use $\frac{6}{8} = \frac{3}{4}$ of the amount of tea. For 12 people, she will need $12 \times \frac{3}{4} = 9$ ounces of tea. Answer: 9

Analysis:
Let's think step by step. Artemis uses 1 ounce of tea for an 8-ounce cup, so for a 6-ounce cup, she will use $\frac{6}{8} = \frac{3}{4}$ of an ounce of tea. For 12 people, she needs $12 \times \frac{3}{4} = 9$ ounces of tea.

Answer:
9

Question:
{User Question}

Your previous answers:
{Previous answers}

Analysis:

---

*Table 12.* Performances of different reasoning methods on Accuracy (%) across all benchmarks using Qwen2-70B-Instruct.

| Method | AIME-2024 | | GSM8K | | GSM-Hard | | SVAMP | | StrategyQA | |
|---|---|---|---|---|---|---|---|---|---|---|
| | Zero Shot | Few Shot | Zero Shot | Few Shot | Zero Shot | Few Shot | Zero Shot | Few Shot | Zero Shot | Few Shot |
| COT | 0 | 3.3 | 91.0 | 91.0 | 51.5 | 65.0 | 93.0 | 92.0 | 79.0 | 90.0 |
| SC($\tau = 0.4$) | $3.3_{\pm2.7}$ | $3.3_{\pm3.3}$ | $93.3_{\pm0.6}$ | $91.8_{\pm1.0}$ | $62.3_{\pm0.6}$ | $68.7_{\pm1.5}$ | $93.2_{\pm0.3}$ | $93.8_{\pm0.3}$ | $78.2_{\pm0.8}$ | $89.6_{\pm1.4}$ |
| SC($\tau = 0.6$) | $2.2_{\pm3.3}$ | $2.2_{\pm3.8}$ | $93.7_{\pm0.3}$ | $92.2_{\pm1.0}$ | $62.7_{\pm0.3}$ | $68.2_{\pm1.4}$ | $93.8_{\pm0.6}$ | $93.1_{\pm0.5}$ | $78.8_{\pm1.6}$ | $90.0_{\pm1.3}$ |
| SC($\tau = 0.8$) | $3.3_{\pm1.9}$ | $2.2_{\pm1.9}$ | $94.0_{\pm0.5}$ | $93.5_{\pm0.5}$ | $62.8_{\pm2.0}$ | $68.3_{\pm0.3}$ | $93.7_{\pm0.3}$ | $93.7_{\pm0.3}$ | $78.4_{\pm2.5}$ | $88.8_{\pm2.3}$ |
| FIRE | $2.2_{\pm1.9}$ | $3.3_{\pm3.8}$ | $91.4_{\pm0.8}$ | $92.5_{\pm0.4}$ | $60.3_{\pm0.6}$ | $65.7_{\pm2.0}$ | $93.5_{\pm0.9}$ | $93.5_{\pm0.5}$ | $78.2_{\pm1.9}$ | $89.5_{\pm0.9}$ |
| CoT-Decoding | $2.2_{\pm1.9}$ | $2.2_{\pm1.9}$ | $93.6_{\pm1.9}$ | $93.5_{\pm1.1}$ | $61.0_{\pm2.3}$ | $66.0_{\pm1.7}$ | $\mathbf{94.2}_{\pm1.2}$ | $93.8_{\pm1.5}$ | $78.8_{\pm1.6}$ | $89.0_{\pm1.5}$ |
| RAP | - | $4.4_{\pm3.4}$ | - | $93.5_{\pm0.4}$ | - | $69.1_{\pm1.9}$ | - | $93.7_{\pm2.1}$ | - | $\mathbf{90.4}_{\pm2.3}$ |
| *Soft Reasoning* | $\mathbf{6.7}_{\pm2.7}$ | $\mathbf{11.1}_{\pm1.7}$ | $\mathbf{94.3}_{\pm0.3}$ | $\mathbf{94.8}_{\pm1.3}$ | $\mathbf{63.3}_{\pm0.6}$ | $\mathbf{72.2}_{\pm0.6}$ | $94.0_{\pm1.0}$ | $\mathbf{94.2}_{\pm1.3}$ | $\mathbf{79.6}_{\pm0.3}$ | $89.2_{\pm1.2}$ |

*Table 13.* Performances of different reasoning methods on Accuracy (%) across all benchmarks.

| Method | LLaMA3-8B-Instruct | | | | | Qwen2-7B-Instruct | | | | | Mistral-7B-Instruct | | | | |
|---|---|---|---|---|---|---|---|---|---|---|---|---|---|---|---|
| | Shot 0 | Shot 1 | Shot 2 | Shot 4 | Shot 8 | Shot 0 | Shot 1 | Shot 2 | Shot 4 | Shot 8 | Shot 0 | Shot 1 | Shot 2 | Shot 4 | Shot 8 |
| | | | | | | *GSM8K* | | | | | | | | | |
| COT | $53.0_{\pm0.0}$ | $73.0_{\pm0.0}$ | $73.5_{\pm0.0}$ | $79.0_{\pm0.0}$ | $77.4_{\pm0.0}$ | $64.5_{\pm0.0}$ | $70.0_{\pm0.0}$ | $66.5_{\pm0.0}$ | $81.5_{\pm0.0}$ | $82.5_{\pm0.0}$ | $42.0_{\pm0.0}$ | $43.5_{\pm0.0}$ | $53.5_{\pm0.0}$ | $54.5_{\pm0.0}$ | $54.0_{\pm0.0}$ |
| SC($\tau=0.4$) | $73.0_{\pm1.6}$ | $75.4_{\pm2.8}$ | $75.7_{\pm2.4}$ | $80.7_{\pm1.1}$ | $80.4_{\pm1.4}$ | $81.2_{\pm0.6}$ | $82.4_{\pm1.8}$ | $83.7_{\pm0.4}$ | $86.9_{\pm0.6}$ | $85.7_{\pm1.5}$ | $52.9_{\pm0.5}$ | $53.5_{\pm1.5}$ | $59.9_{\pm2.3}$ | $60.4_{\pm1.7}$ | $58.3_{\pm1.5}$ |
| SC($\tau=0.6$) | $73.6_{\pm2.5}$ | $73.4_{\pm3.6}$ | $72.6_{\pm1.9}$ | $80.3_{\pm0.9}$ | $80.6_{\pm1.5}$ | $80.2_{\pm1.9}$ | $84.5_{\pm2.1}$ | $84.2_{\pm0.4}$ | $86.5_{\pm0.9}$ | $85.4_{\pm0.9}$ | $55.1_{\pm3.6}$ | $56.5_{\pm2.0}$ | $59.6_{\pm0.7}$ | $60.5_{\pm1.2}$ | $57.4_{\pm1.0}$ |
| SC($\tau=0.8$) | $65.0_{\pm2.0}$ | $74.5_{\pm0.7}$ | $66.8_{\pm1.7}$ | $80.9_{\pm1.5}$ | $81.1_{\pm1.1}$ | $80.0_{\pm0.9}$ | $82.7_{\pm2.9}$ | $82.6_{\pm0.7}$ | $85.4_{\pm0.8}$ | $85.1_{\pm1.6}$ | $50.2_{\pm2.6}$ | $51.4_{\pm1.4}$ | $56.4_{\pm2.7}$ | $59.8_{\pm2.6}$ | $57.7_{\pm2.6}$ |
| FIRE | $73.8_{\pm2.3}$ | $75.4_{\pm1.6}$ | $76.4_{\pm2.0}$ | $78.4_{\pm3.2}$ | $79.6_{\pm2.9}$ | $81.0_{\pm1.9}$ | $81.0_{\pm2.9}$ | $73.6_{\pm2.2}$ | $82.5_{\pm2.1}$ | $83.0_{\pm1.3}$ | $47.2_{\pm2.9}$ | $52.4_{\pm2.3}$ | $53.8_{\pm1.7}$ | $60.6_{\pm0.9}$ | $56.1_{\pm3.2}$ |
| CoT-Decoding | $73.9_{\pm1.9}$ | $76.7_{\pm1.4}$ | $78.6_{\pm1.7}$ | $81.4_{\pm1.8}$ | $80.3_{\pm1.7}$ | $82.0_{\pm2.8}$ | $81.6_{\pm2.5}$ | $76.2_{\pm2.5}$ | $85.7_{\pm0.8}$ | $84.5_{\pm2.1}$ | $47.3_{\pm3.0}$ | $56.4_{\pm3.3}$ | $57.3_{\pm2.5}$ | $57.6_{\pm3.4}$ | $58.2_{\pm2.3}$ |
| RAP | - | $77.4_{\pm1.7}$ | $79.4_{\pm1.6}$ | $80.4_{\pm1.4}$ | $80.7_{\pm1.4}$ | - | $84.7_{\pm2.0}$ | $85.7_{\pm1.0}$ | $87.4_{\pm0.9}$ | $86.2_{\pm1.2}$ | - | $57.6_{\pm1.8}$ | $57.1_{\pm1.7}$ | $59.8_{\pm1.6}$ | $58.6_{\pm1.8}$ |
| *Soft Reasoning* | $\mathbf{79.4}_{\pm1.2}$ | $\mathbf{79.4}_{\pm2.8}$ | $\mathbf{83.0}_{\pm0.8}$ | $\mathbf{83.5}_{\pm0.9}$ | $\mathbf{84.3}_{\pm1.4}$ | $\mathbf{88.6}_{\pm1.2}$ | $\mathbf{88.8}_{\pm1.4}$ | $\mathbf{88.4}_{\pm1.0}$ | $\mathbf{89.8}_{\pm2.5}$ | $\mathbf{90.0}_{\pm1.4}$ | $\mathbf{61.4}_{\pm2.5}$ | $\mathbf{61.2}_{\pm2.5}$ | $\mathbf{61.2}_{\pm0.9}$ | $\mathbf{61.4}_{\pm1.9}$ | $\mathbf{62.7}_{\pm1.0}$ |
| | | | | | | *GSM-Hard* | | | | | | | | | |
| COT | $14.0_{\pm0.0}$ | $22.5_{\pm0.0}$ | $24.5_{\pm0.0}$ | $26.5_{\pm0.0}$ | $28.0_{\pm0.0}$ | $40.0_{\pm0.0}$ | $39.0_{\pm0.0}$ | $48.0_{\pm0.0}$ | $53.0_{\pm0.0}$ | $55.5_{\pm0.0}$ | $14.5_{\pm0.0}$ | $22.0_{\pm0.0}$ | $21.5_{\pm0.0}$ | $23.5_{\pm0.0}$ | $24.0_{\pm0.0}$ |
| SC($\tau=0.4$) | $25.7_{\pm0.4}$ | $25.3_{\pm0.4}$ | $28.6_{\pm1.2}$ | $32.2_{\pm0.4}$ | $31.8_{\pm1.8}$ | $47.5_{\pm1.4}$ | $48.6_{\pm1.4}$ | $53.7_{\pm1.3}$ | $55.2_{\pm1.5}$ | $55.4_{\pm0.7}$ | $19.5_{\pm1.0}$ | $26.5_{\pm1.6}$ | $26.9_{\pm1.4}$ | $27.3_{\pm1.0}$ | $26.1_{\pm1.5}$ |
| SC($\tau=0.6$) | $24.5_{\pm1.1}$ | $25.7_{\pm0.5}$ | $28.4_{\pm1.4}$ | $32.0_{\pm1.1}$ | $31.2_{\pm1.3}$ | $46.2_{\pm1.9}$ | $50.1_{\pm2.9}$ | $54.2_{\pm0.7}$ | $56.0_{\pm1.1}$ | $53.4_{\pm0.6}$ | $20.7_{\pm1.5}$ | $25.7_{\pm0.9}$ | $27.8_{\pm1.2}$ | $31.0_{\pm2.5}$ | $25.3_{\pm1.6}$ |
| SC($\tau=0.8$) | $21.8_{\pm1.3}$ | $24.8_{\pm1.9}$ | $28.2_{\pm2.7}$ | $30.6_{\pm1.2}$ | $30.8_{\pm0.9}$ | $47.3_{\pm1.3}$ | $47.8_{\pm2.7}$ | $54.8_{\pm1.5}$ | $55.7_{\pm1.3}$ | $55.4_{\pm0.9}$ | $19.1_{\pm2.0}$ | $26.8_{\pm2.5}$ | $26.6_{\pm1.2}$ | $29.8_{\pm2.1}$ | $26.6_{\pm1.1}$ |
| FIRE | $25.2_{\pm3.0}$ | $24.1_{\pm1.5}$ | $27.0_{\pm1.5}$ | $27.9_{\pm1.8}$ | $25.7_{\pm2.1}$ | $45.1_{\pm2.0}$ | $43.4_{\pm2.5}$ | $52.9_{\pm1.9}$ | $49.0_{\pm2.5}$ | $51.0_{\pm1.8}$ | $18.1_{\pm1.9}$ | $25.8_{\pm1.3}$ | $27.0_{\pm1.7}$ | $25.5_{\pm2.8}$ | $26.3_{\pm1.4}$ |
| CoT-Decoding | $24.8_{\pm1.3}$ | $27.3_{\pm1.6}$ | $28.0_{\pm0.4}$ | $31.0_{\pm1.8}$ | $30.3_{\pm1.3}$ | $46.7_{\pm2.3}$ | $44.1_{\pm1.8}$ | $53.9_{\pm1.5}$ | $50.1_{\pm1.2}$ | $52.1_{\pm1.0}$ | $16.6_{\pm0.7}$ | $26.6_{\pm2.6}$ | $28.0_{\pm0.7}$ | $26.0_{\pm2.0}$ | $27.4_{\pm1.6}$ |
| RAP | - | $26.7_{\pm1.2}$ | $31.4_{\pm1.2}$ | $32.4_{\pm1.1}$ | $32.7_{\pm1.2}$ | - | $53.2_{\pm1.9}$ | $55.1_{\pm1.2}$ | $55.9_{\pm1.3}$ | $56.2_{\pm0.8}$ | - | $27.4_{\pm1.5}$ | $28.0_{\pm1.0}$ | $28.4_{\pm1.7}$ | $27.6_{\pm1.2}$ |
| *Soft Reasoning* | $\mathbf{28.2}_{\pm1.8}$ | $\mathbf{30.2}_{\pm1.2}$ | $\mathbf{35.3}_{\pm1.4}$ | $\mathbf{33.2}_{\pm0.6}$ | $\mathbf{35.7}_{\pm1.0}$ | $\mathbf{53.7}_{\pm1.6}$ | $\mathbf{57.4}_{\pm0.7}$ | $\mathbf{57.5}_{\pm1.0}$ | $\mathbf{57.4}_{\pm1.2}$ | $\mathbf{58.7}_{\pm0.5}$ | $\mathbf{25.8}_{\pm1.8}$ | $\mathbf{29.1}_{\pm1.2}$ | $\mathbf{32.3}_{\pm0.4}$ | $\mathbf{30.0}_{\pm1.2}$ | $\mathbf{32.5}_{\pm1.5}$ |
| | | | | | | *SVAMP* | | | | | | | | | |
| COT | $61.0_{\pm0.0}$ | $81.0_{\pm0.0}$ | $83.5_{\pm0.0}$ | $84.0_{\pm0.0}$ | $83.0_{\pm0.0}$ | $43.5_{\pm0.0}$ | $83.0_{\pm0.0}$ | $84.0_{\pm0.0}$ | $85.5_{\pm0.0}$ | $86.0_{\pm0.0}$ | $52.0_{\pm0.0}$ | $65.0_{\pm0.0}$ | $66.0_{\pm0.0}$ | $69.5_{\pm0.0}$ | $72.0_{\pm0.0}$ |
| SC($\tau=0.4$) | $79.1_{\pm1.2}$ | $85.8_{\pm1.5}$ | $86.5_{\pm0.7}$ | $87.3_{\pm0.8}$ | $87.1_{\pm1.0}$ | $72.3_{\pm2.0}$ | $90.2_{\pm1.0}$ | $89.4_{\pm0.5}$ | $90.3_{\pm0.9}$ | $90.3_{\pm1.2}$ | $67.4_{\pm2.5}$ | $74.5_{\pm1.7}$ | $73.7_{\pm1.0}$ | $76.5_{\pm1.5}$ | $77.8_{\pm1.0}$ |
| SC($\tau=0.6$) | $76.1_{\pm3.9}$ | $86.2_{\pm0.5}$ | $86.8_{\pm1.7}$ | $86.9_{\pm1.2}$ | $87.7_{\pm1.2}$ | $77.3_{\pm1.2}$ | $90.6_{\pm0.9}$ | $90.2_{\pm0.8}$ | $91.4_{\pm0.9}$ | $90.4_{\pm0.6}$ | $69.7_{\pm1.6}$ | $75.8_{\pm1.5}$ | $75.6_{\pm0.8}$ | $75.8_{\pm1.4}$ | $78.4_{\pm2.0}$ |
| SC($\tau=0.8$) | $69.6_{\pm2.0}$ | $86.3_{\pm2.2}$ | $86.9_{\pm1.0}$ | $87.4_{\pm1.5}$ | $87.4_{\pm1.2}$ | $78.6_{\pm2.1}$ | $90.3_{\pm0.7}$ | $90.1_{\pm1.0}$ | $90.8_{\pm0.7}$ | $90.6_{\pm1.2}$ | $68.3_{\pm0.9}$ | $75.1_{\pm0.7}$ | $76.6_{\pm1.5}$ | $76.9_{\pm1.6}$ | $77.6_{\pm1.1}$ |
| FIRE | $81.5_{\pm0.8}$ | $86.6_{\pm1.8}$ | $86.1_{\pm1.3}$ | $86.1_{\pm1.4}$ | $87.6_{\pm2.0}$ | $76.3_{\pm2.2}$ | $89.9_{\pm1.4}$ | $89.7_{\pm0.8}$ | $89.3_{\pm0.9}$ | $90.6_{\pm0.2}$ | $67.1_{\pm1.9}$ | $77.7_{\pm1.1}$ | $76.9_{\pm2.7}$ | $77.8_{\pm1.2}$ | $78.6_{\pm1.4}$ |
| CoT-Decoding | $83.2_{\pm1.2}$ | $87.8_{\pm1.0}$ | $87.5_{\pm1.0}$ | $87.5_{\pm1.3}$ | $88.2_{\pm1.0}$ | $78.6_{\pm1.6}$ | $90.3_{\pm0.4}$ | $90.0_{\pm1.0}$ | $90.3_{\pm1.0}$ | $89.7_{\pm0.5}$ | $69.4_{\pm2.5}$ | $77.8_{\pm2.0}$ | $77.7_{\pm1.5}$ | $76.9_{\pm2.5}$ | $78.6_{\pm1.4}$ |
| RAP | - | $78.4_{\pm1.2}$ | $87.4_{\pm1.0}$ | $86.8_{\pm1.0}$ | $87.9_{\pm1.1}$ | - | $90.8_{\pm0.7}$ | $91.2_{\pm0.7}$ | $90.1_{\pm0.7}$ | $90.8_{\pm0.6}$ | - | $0.0_{\pm1.2}$ | $0.0_{\pm1.3}$ | $78.4_{\pm1.4}$ | $79.4_{\pm1.1}$ |
| *Soft Reasoning* | $\mathbf{88.2}_{\pm1.3}$ | $\mathbf{89.2}_{\pm1.5}$ | $\mathbf{88.8}_{\pm1.0}$ | $\mathbf{89.1}_{\pm0.8}$ | $\mathbf{90.2}_{\pm0.6}$ | $\mathbf{83.4}_{\pm2.4}$ | $\mathbf{92.3}_{\pm0.8}$ | $\mathbf{92.4}_{\pm0.8}$ | $\mathbf{90.8}_{\pm0.8}$ | $\mathbf{92.2}_{\pm0.8}$ | $\mathbf{72.2}_{\pm2.2}$ | $\mathbf{79.0}_{\pm0.9}$ | $\mathbf{78.4}_{\pm1.3}$ | $\mathbf{80.0}_{\pm1.4}$ | $\mathbf{82.1}_{\pm1.2}$ |
| | | | | | | *StrategyQA* | | | | | | | | | |
| COT | $58.5_{\pm0.0}$ | $63.0_{\pm0.0}$ | $68.0_{\pm0.0}$ | $67.5_{\pm0.0}$ | $68.5_{\pm0.0}$ | $63.0_{\pm0.0}$ | $54.5_{\pm0.0}$ | $63.5_{\pm0.0}$ | $66.0_{\pm0.0}$ | $70.0_{\pm0.0}$ | $62.0_{\pm0.0}$ | $62.5_{\pm0.0}$ | $63.5_{\pm0.0}$ | $68.5_{\pm0.0}$ | $69.0_{\pm0.0}$ |
| SC($\tau=0.4$) | $64.7_{\pm0.7}$ | $68.4_{\pm2.4}$ | $68.6_{\pm1.7}$ | $69.2_{\pm1.2}$ | $71.6_{\pm0.8}$ | $67.1_{\pm1.5}$ | $67.4_{\pm2.0}$ | $66.5_{\pm1.2}$ | $69.1_{\pm1.2}$ | $71.1_{\pm1.6}$ | $63.9_{\pm1.5}$ | $57.6_{\pm2.0}$ | $65.9_{\pm0.7}$ | $70.2_{\pm1.9}$ | $72.6_{\pm1.2}$ |
| SC($\tau=0.6$) | $59.9_{\pm2.0}$ | $69.9_{\pm1.2}$ | $68.2_{\pm0.8}$ | $70.7_{\pm2.2}$ | $71.3_{\pm1.5}$ | $67.5_{\pm0.7}$ | $66.5_{\pm1.7}$ | $67.4_{\pm2.6}$ | $67.7_{\pm1.4}$ | $70.1_{\pm0.8}$ | $64.9_{\pm1.0}$ | $58.7_{\pm1.3}$ | $64.8_{\pm0.7}$ | $69.9_{\pm1.4}$ | $71.7_{\pm0.8}$ |
| SC($\tau=0.8$) | $54.4_{\pm2.6}$ | $68.0_{\pm2.0}$ | $67.9_{\pm0.7}$ | $70.4_{\pm0.9}$ | $72.7_{\pm1.2}$ | $67.0_{\pm1.0}$ | $68.0_{\pm1.9}$ | $68.3_{\pm1.2}$ | $67.7_{\pm2.5}$ | $70.1_{\pm0.8}$ | $64.9_{\pm1.0}$ | $59.9_{\pm1.2}$ | $66.5_{\pm0.4}$ | $69.9_{\pm1.4}$ | $72.1_{\pm1.5}$ |
| FIRE | $63.0_{\pm3.7}$ | $70.8_{\pm1.6}$ | $71.8_{\pm1.3}$ | $70.2_{\pm2.0}$ | $72.8_{\pm1.5}$ | $67.6_{\pm0.8}$ | $68.4_{\pm0.8}$ | $68.4_{\pm1.8}$ | $67.3_{\pm1.2}$ | $68.1_{\pm0.8}$ | $64.2_{\pm1.0}$ | $66.5_{\pm1.5}$ | $68.2_{\pm1.4}$ | $72.6_{\pm2.1}$ | $71.0_{\pm2.2}$ |
| CoT-Decoding | $64.6_{\pm1.6}$ | $71.1_{\pm3.1}$ | $71.4_{\pm2.0}$ | $70.5_{\pm2.0}$ | $73.3_{\pm1.8}$ | $65.9_{\pm1.5}$ | $68.3_{\pm2.5}$ | $68.7_{\pm0.7}$ | $67.5_{\pm1.5}$ | $69.5_{\pm2.1}$ | $63.5_{\pm1.5}$ | $68.2_{\pm0.4}$ | $68.6_{\pm0.7}$ | $70.6_{\pm1.6}$ | $72.7_{\pm2.1}$ |
| RAP | - | $71.6_{\pm1.7}$ | $70.6_{\pm1.1}$ | $71.5_{\pm1.4}$ | $73.4_{\pm1.1}$ | - | $67.5_{\pm1.5}$ | $68.2_{\pm1.3}$ | $68.5_{\pm1.2}$ | $\mathbf{71.3}_{\pm1.1}$ | - | $68.5_{\pm1.1}$ | $70.6_{\pm0.7}$ | $72.1_{\pm1.4}$ | $72.4_{\pm1.3}$ |
| *Soft Reasoning* | $\mathbf{67.2}_{\pm0.7}$ | $71.0_{\pm0.9}$ | $\mathbf{72.3}_{\pm1.2}$ | $\mathbf{73.8}_{\pm1.2}$ | $\mathbf{75.6}_{\pm0.8}$ | $\mathbf{68.1}_{\pm1.5}$ | $\mathbf{69.6}_{\pm1.6}$ | $\mathbf{69.7}_{\pm1.2}$ | $\mathbf{68.9}_{\pm0.5}$ | $70.3_{\pm1.3}$ | $\mathbf{66.1}_{\pm1.9}$ | $\mathbf{70.1}_{\pm0.7}$ | $\mathbf{71.2}_{\pm1.4}$ | $\mathbf{72.7}_{\pm1.0}$ | $\mathbf{72.8}_{\pm1.5}$ |

*Table 14.* Coverage rates of correct answers across different models (%) on all benchmarks.

| Method | LLaMA3-8B-Instruct | | | | | Qwen2-7B-Instruct | | | | | Mistral-7B-Instruct | | | | |
|---|---|---|---|---|---|---|---|---|---|---|---|---|---|---|---|
| | Shot 0 | Shot 1 | Shot 2 | Shot 4 | Shot 8 | Shot 0 | Shot 1 | Shot 2 | Shot 4 | Shot 8 | Shot 0 | Shot 1 | Shot 2 | Shot 4 | Shot 8 |
| | | | | | | *GSM8K* | | | | | | | | | |
| SC($\tau=0.4$) | $79.8_{\pm0.8}$ | $69.2_{\pm2.8}$ | $75.7_{\pm2.4}$ | $89.5_{\pm1.4}$ | $91.0_{\pm1.5}$ | $93.1_{\pm0.4}$ | $88.9_{\pm1.7}$ | $91.4_{\pm1.0}$ | $93.5_{\pm0.7}$ | $94.5_{\pm0.6}$ | $73.2_{\pm3.0}$ | $69.3_{\pm1.6}$ | $75.5_{\pm2.2}$ | $74.9_{\pm1.9}$ | $73.4_{\pm1.0}$ |
| SC($\tau=0.6$) | $78.4_{\pm2.2}$ | $66.6_{\pm3.6}$ | $72.6_{\pm1.9}$ | $90.6_{\pm1.3}$ | $91.1_{\pm0.8}$ | $94.2_{\pm0.7}$ | $88.5_{\pm1.5}$ | $92.4_{\pm0.7}$ | $93.8_{\pm0.7}$ | $94.2_{\pm1.3}$ | $73.0_{\pm1.3}$ | $72.7_{\pm2.7}$ | $76.2_{\pm1.6}$ | $76.5_{\pm2.5}$ | $73.5_{\pm2.0}$ |
| SC($\tau=0.8$) | $71.0_{\pm1.2}$ | $62.1_{\pm0.7}$ | $66.8_{\pm1.7}$ | $90.4_{\pm1.6}$ | $90.2_{\pm1.6}$ | $92.9_{\pm1.2}$ | $88.6_{\pm1.5}$ | $91.6_{\pm1.1}$ | $93.1_{\pm0.9}$ | $93.8_{\pm0.7}$ | $71.2_{\pm1.8}$ | $70.1_{\pm2.0}$ | $74.3_{\pm2.8}$ | $74.5_{\pm2.8}$ | $73.2_{\pm1.4}$ |
| FIRE | $84.5_{\pm1.3}$ | $83.9_{\pm2.0}$ | $88.8_{\pm2.0}$ | $89.4_{\pm1.1}$ | $90.5_{\pm0.9}$ | $86.8_{\pm1.8}$ | $78.1_{\pm2.6}$ | $84.2_{\pm1.3}$ | $90.6_{\pm1.9}$ | $90.8_{\pm0.9}$ | $63.1_{\pm2.5}$ | $70.4_{\pm2.5}$ | $70.9_{\pm1.6}$ | $76.7_{\pm2.2}$ | $73.9_{\pm2.8}$ |
| CoT-Decoding | $85.3_{\pm1.7}$ | $88.1_{\pm1.2}$ | $90.6_{\pm1.1}$ | $90.2_{\pm0.8}$ | $90.5_{\pm0.9}$ | $88.4_{\pm2.1}$ | $81.2_{\pm2.0}$ | $85.5_{\pm2.4}$ | $92.0_{\pm0.9}$ | $91.7_{\pm0.8}$ | $63.2_{\pm2.4}$ | $72.6_{\pm2.7}$ | $75.5_{\pm2.5}$ | $74.3_{\pm2.3}$ | $76.6_{\pm1.2}$ |
| RAP | - | $87.6_{\pm0.9}$ | $89.1_{\pm1.3}$ | $88.9_{\pm1.2}$ | $89.5_{\pm0.8}$ | - | $88.3_{\pm1.2}$ | $92.4_{\pm1.2}$ | $93.4_{\pm1.1}$ | $92.3_{\pm0.7}$ | - | $72.4_{\pm2.1}$ | $74.5_{\pm1.3}$ | $75.8_{\pm1.5}$ | $75.6_{\pm1.3}$ |
| *Soft Reasoning* | $\mathbf{91.8}_{\pm1.4}$ | $\mathbf{91.1}_{\pm0.8}$ | $\mathbf{92.4}_{\pm1.1}$ | $\mathbf{92.6}_{\pm1.4}$ | $\mathbf{92.2}_{\pm0.8}$ | $\mathbf{95.9}_{\pm0.1}$ | $\mathbf{96.8}_{\pm0.6}$ | $\mathbf{96.4}_{\pm1.0}$ | $\mathbf{96.8}_{\pm1.4}$ | $\mathbf{96.6}_{\pm0.7}$ | $\mathbf{85.4}_{\pm1.4}$ | $\mathbf{79.0}_{\pm2.4}$ | $\mathbf{82.3}_{\pm1.0}$ | $\mathbf{81.9}_{\pm1.4}$ | $\mathbf{82.5}_{\pm0.9}$ |
| | | | | | | *GSM-Hard* | | | | | | | | | |
| SC($\tau=0.4$) | $27.3_{\pm0.4}$ | $31.7_{\pm0.5}$ | $38.6_{\pm1.9}$ | $41.6_{\pm1.8}$ | $43.4_{\pm0.4}$ | $62.6_{\pm1.6}$ | $53.4_{\pm1.7}$ | $62.4_{\pm1.2}$ | $62.6_{\pm1.7}$ | $64.3_{\pm0.7}$ | $31.1_{\pm1.1}$ | $37.5_{\pm1.6}$ | $40.9_{\pm1.2}$ | $39.9_{\pm1.6}$ | $38.2_{\pm1.5}$ |
| SC($\tau=0.6$) | $28.2_{\pm1.6}$ | $33.0_{\pm0.8}$ | $38.0_{\pm1.4}$ | $39.9_{\pm2.2}$ | $43.0_{\pm1.1}$ | $63.3_{\pm2.0}$ | $56.2_{\pm2.2}$ | $62.4_{\pm1.1}$ | $65.3_{\pm0.7}$ | $65.8_{\pm1.4}$ | $30.2_{\pm1.7}$ | $37.3_{\pm0.9}$ | $38.8_{\pm1.3}$ | $41.3_{\pm1.4}$ | $39.1_{\pm2.5}$ |
| SC($\tau=0.8$) | $25.1_{\pm0.6}$ | $32.3_{\pm2.4}$ | $37.8_{\pm1.2}$ | $39.3_{\pm1.2}$ | $42.2_{\pm1.1}$ | $61.8_{\pm0.5}$ | $54.9_{\pm1.9}$ | $64.1_{\pm1.0}$ | $65.1_{\pm1.0}$ | $65.1_{\pm1.1}$ | $31.7_{\pm1.4}$ | $38.2_{\pm1.6}$ | $39.2_{\pm1.0}$ | $40.9_{\pm1.8}$ | $37.0_{\pm1.1}$ |
| FIRE | $32.0_{\pm2.3}$ | $33.1_{\pm2.5}$ | $38.3_{\pm1.4}$ | $39.9_{\pm1.1}$ | $40.6_{\pm2.0}$ | $54.4_{\pm2.1}$ | $49.3_{\pm2.3}$ | $56.7_{\pm2.1}$ | $56.8_{\pm2.1}$ | $60.6_{\pm1.2}$ | $26.2_{\pm2.5}$ | $37.4_{\pm2.1}$ | $39.8_{\pm1.8}$ | $39.1_{\pm2.7}$ | $35.8_{\pm1.6}$ |
| CoT-Decoding | $33.1_{\pm0.9}$ | $33.6_{\pm1.9}$ | $39.8_{\pm1.4}$ | $42.3_{\pm1.2}$ | $42.2_{\pm1.5}$ | $55.1_{\pm1.0}$ | $49.3_{\pm2.2}$ | $56.6_{\pm1.3}$ | $57.4_{\pm0.6}$ | $61.5_{\pm1.4}$ | $27.7_{\pm1.2}$ | $36.9_{\pm1.0}$ | $40.3_{\pm1.6}$ | $38.5_{\pm1.4}$ | $38.4_{\pm0.7}$ |
| RAP | - | $33.4_{\pm1.6}$ | $40.2_{\pm1.5}$ | $41.9_{\pm1.0}$ | $43.9_{\pm1.3}$ | - | $53.7_{\pm1.8}$ | $60.2_{\pm1.2}$ | $62.1_{\pm1.1}$ | $64.2_{\pm1.2}$ | - | $37.6_{\pm1.5}$ | $40.6_{\pm1.2}$ | $40.3_{\pm1.7}$ | $38.4_{\pm1.4}$ |
| *Soft Reasoning* | $\mathbf{37.0}_{\pm1.5}$ | $\mathbf{37.0}_{\pm0.8}$ | $\mathbf{46.4}_{\pm2.5}$ | $\mathbf{47.3}_{\pm2.2}$ | $\mathbf{49.8}_{\pm1.0}$ | $\mathbf{69.4}_{\pm1.7}$ | $\mathbf{67.8}_{\pm0.9}$ | $\mathbf{70.7}_{\pm1.0}$ | $\mathbf{71.1}_{\pm1.1}$ | $\mathbf{71.1}_{\pm1.1}$ | $\mathbf{40.7}_{\pm1.2}$ | $\mathbf{43.3}_{\pm1.2}$ | $\mathbf{44.4}_{\pm1.5}$ | $\mathbf{45.9}_{\pm1.1}$ | $\mathbf{45.2}_{\pm1.9}$ |
| | | | | | | *SVAMP* | | | | | | | | | |
| SC($\tau=0.4$) | $84.6_{\pm1.1}$ | $91.8_{\pm0.5}$ | $91.6_{\pm1.6}$ | $92.8_{\pm0.4}$ | $92.9_{\pm0.6}$ | $91.7_{\pm1.8}$ | $94.3_{\pm0.6}$ | $93.9_{\pm0.7}$ | $93.7_{\pm0.4}$ | $94.0_{\pm0.9}$ | $61.9_{\pm1.7}$ | $85.2_{\pm1.3}$ | $84.5_{\pm1.1}$ | $86.3_{\pm1.3}$ | $87.7_{\pm0.4}$ |
| SC($\tau=0.6$) | $82.1_{\pm2.7}$ | $92.3_{\pm0.2}$ | $92.7_{\pm1.0}$ | $93.0_{\pm0.6}$ | $93.8_{\pm0.4}$ | $92.3_{\pm1.5}$ | $94.4_{\pm0.6}$ | $94.6_{\pm0.6}$ | $94.3_{\pm0.5}$ | $94.0_{\pm1.0}$ | $67.5_{\pm1.4}$ | $85.7_{\pm0.5}$ | $86.3_{\pm1.2}$ | $87.9_{\pm1.4}$ | $88.9_{\pm1.4}$ |
| SC($\tau=0.8$) | $73.8_{\pm2.6}$ | $91.7_{\pm1.7}$ | $93.3_{\pm0.5}$ | $93.5_{\pm0.8}$ | $94.6_{\pm0.7}$ | $92.5_{\pm1.1}$ | $94.5_{\pm0.7}$ | $94.5_{\pm0.5}$ | $94.1_{\pm0.9}$ | $94.0_{\pm0.0}$ | $72.1_{\pm2.3}$ | $86.5_{\pm0.8}$ | $87.1_{\pm1.1}$ | $88.3_{\pm1.0}$ | $88.2_{\pm1.2}$ |
| FIRE | $89.9_{\pm0.5}$ | $93.7_{\pm0.3}$ | $93.5_{\pm0.9}$ | $92.9_{\pm1.2}$ | $93.8_{\pm1.3}$ | $91.5_{\pm0.5}$ | $93.9_{\pm1.1}$ | $94.7_{\pm1.0}$ | $94.5_{\pm1.8}$ | $95.1_{\pm0.7}$ | $68.2_{\pm1.4}$ | $87.9_{\pm1.3}$ | $89.9_{\pm1.7}$ | $90.2_{\pm1.5}$ | $90.1_{\pm1.6}$ |
| CoT-Decoding | $90.8_{\pm0.8}$ | $93.5_{\pm1.2}$ | $94.2_{\pm0.8}$ | $93.8_{\pm0.8}$ | $93.9_{\pm1.3}$ | $91.2_{\pm1.0}$ | $94.4_{\pm0.5}$ | $94.2_{\pm0.7}$ | $94.6_{\pm1.0}$ | $93.4_{\pm0.9}$ | $67.0_{\pm2.9}$ | $89.5_{\pm0.5}$ | $88.7_{\pm1.3}$ | $88.9_{\pm1.7}$ | $90.4_{\pm1.2}$ |
| RAP | - | $93.6_{\pm1.1}$ | $93.8_{\pm0.8}$ | $93.4_{\pm0.9}$ | $93.9_{\pm1.3}$ | - | $94.3_{\pm1.1}$ | $94.7_{\pm0.9}$ | $94.8_{\pm0.7}$ | $94.8_{\pm0.7}$ | - | $88.7_{\pm1.2}$ | $89.4_{\pm1.1}$ | $89.7_{\pm1.3}$ | $90.7_{\pm1.0}$ |
| *Soft Reasoning* | $\mathbf{93.8}_{\pm1.4}$ | $\mathbf{95.5}_{\pm0.9}$ | $\mathbf{95.5}_{\pm0.8}$ | $\mathbf{95.1}_{\pm0.7}$ | $\mathbf{95.8}_{\pm1.2}$ | $\mathbf{97.0}_{\pm0.7}$ | $\mathbf{96.8}_{\pm0.3}$ | $\mathbf{95.6}_{\pm0.8}$ | $\mathbf{96.0}_{\pm1.2}$ | $\mathbf{97.8}_{\pm0.5}$ | $\mathbf{78.1}_{\pm0.8}$ | $\mathbf{91.2}_{\pm1.7}$ | $\mathbf{90.5}_{\pm1.4}$ | $\mathbf{90.6}_{\pm1.4}$ | $\mathbf{91.0}_{\pm0.6}$ |
| | | | | | | *StrategyQA* | | | | | | | | | |
| SC($\tau=0.4$) | $85.3_{\pm0.2}$ | $85.8_{\pm1.5}$ | $84.7_{\pm1.1}$ | $84.0_{\pm0.3}$ | $85.7_{\pm1.0}$ | $83.5_{\pm1.3}$ | $84.7_{\pm0.8}$ | $85.7_{\pm1.1}$ | $85.0_{\pm1.7}$ | $86.7_{\pm1.7}$ | $73.8_{\pm1.6}$ | $70.5_{\pm1.1}$ | $74.9_{\pm0.6}$ | $84.4_{\pm0.9}$ | $85.4_{\pm1.9}$ |
| SC($\tau=0.6$) | $84.1_{\pm1.7}$ | $89.2_{\pm2.0}$ | $86.3_{\pm2.0}$ | $87.1_{\pm1.3}$ | $88.5_{\pm1.3}$ | $83.2_{\pm1.4}$ | $85.4_{\pm1.4}$ | $86.4_{\pm2.1}$ | $85.9_{\pm1.4}$ | $87.5_{\pm1.3}$ | $75.0_{\pm0.8}$ | $73.6_{\pm1.2}$ | $76.9_{\pm1.0}$ | $86.1_{\pm1.0}$ | $88.0_{\pm1.0}$ |
| SC($\tau=0.8$) | $86.6_{\pm2.9}$ | $88.9_{\pm2.7}$ | $87.1_{\pm0.9}$ | $88.1_{\pm0.9}$ | $89.6_{\pm1.2}$ | $84.8_{\pm1.7}$ | $84.9_{\pm2.8}$ | $86.9_{\pm1.3}$ | $85.2_{\pm1.2}$ | $88.2_{\pm1.5}$ | $76.0_{\pm1.4}$ | $74.1_{\pm0.9}$ | $77.2_{\pm1.6}$ | $87.8_{\pm1.2}$ | $89.8_{\pm0.8}$ |
| FIRE | $91.2_{\pm1.5}$ | $92.6_{\pm1.0}$ | $90.1_{\pm2.1}$ | $89.1_{\pm2.1}$ | $90.6_{\pm1.1}$ | $84.4_{\pm1.6}$ | $87.0_{\pm1.7}$ | $88.4_{\pm2.2}$ | $88.1_{\pm2.3}$ | $89.4_{\pm1.4}$ | $74.9_{\pm1.6}$ | $81.0_{\pm1.0}$ | $79.6_{\pm1.4}$ | $86.4_{\pm1.7}$ | $89.7_{\pm1.5}$ |
| CoT-Decoding | $92.4_{\pm0.4}$ | $93.4_{\pm1.1}$ | $87.4_{\pm1.4}$ | $89.1_{\pm0.6}$ | $90.6_{\pm1.1}$ | $84.7_{\pm2.3}$ | $87.3_{\pm1.3}$ | $86.9_{\pm1.1}$ | $86.8_{\pm0.8}$ | $88.9_{\pm2.4}$ | $75.6_{\pm0.8}$ | $82.2_{\pm2.8}$ | $79.7_{\pm1.4}$ | $87.3_{\pm1.4}$ | $89.0_{\pm1.3}$ |
| RAP | - | $93.5_{\pm1.2}$ | $89.6_{\pm1.4}$ | $89.3_{\pm1.5}$ | $91.1_{\pm1.3}$ | - | $88.6_{\pm1.4}$ | $87.6_{\pm0.9}$ | $88.6_{\pm1.2}$ | $88.7_{\pm1.6}$ | - | $81.6_{\pm1.5}$ | $79.2_{\pm1.3}$ | $86.6_{\pm1.2}$ | $89.3_{\pm1.2}$ |
| *Soft Reasoning* | $\mathbf{93.7}_{\pm1.4}$ | $\mathbf{94.0}_{\pm1.3}$ | $\mathbf{93.5}_{\pm1.2}$ | $\mathbf{90.9}_{\pm1.3}$ | $\mathbf{93.3}_{\pm1.8}$ | $\mathbf{88.4}_{\pm0.7}$ | $\mathbf{89.0}_{\pm0.8}$ | $\mathbf{89.8}_{\pm0.8}$ | $\mathbf{91.6}_{\pm0.8}$ | $\mathbf{90.4}_{\pm1.0}$ | $\mathbf{81.2}_{\pm1.3}$ | $\mathbf{84.1}_{\pm0.7}$ | $\mathbf{82.4}_{\pm1.6}$ | $\mathbf{87.3}_{\pm0.9}$ | $\mathbf{90.2}_{\pm1.4}$ |

