# OpenReview forum: "Soft Reasoning: Navigating Solution Spaces in Large Language Models through Controlled Embedding Exploration"
_ICML.cc/2025/Conference — ICML 2025 spotlightposter_

### Official Review · Reviewer_W5Wg · 2025-03-13

**Overall Recommendation:** 3

**Summary:**

The authors proposed a new LLM inference sampling framework that searches for different reasoning paths by controling a Gaussian embedding that is inserted to the sequence. They proposed embedding perturbation for controlling the sampling in the continuous space, and come up with Bayesian optimisation to guide the sampling via a verifier-guided objective. Comprehensive comparisons with other sampling strategies and ablation studies are provided for proving the effectiveness and efficiency of the framework.

**Claims And Evidence:**

Most claims are well supported by ablation studies or prior works, but not with theoretical provement.

A theoretical provement of "searching for a start token in the continuous embedding space" being better than "sampling every token in the discrete token space" can further improve the soundness of the paper.

**Essential References Not Discussed:**

No.

**Experimental Designs Or Analyses:**

The experiments are comprehensive and convincing. The main experiment includes 3 LLMs and 4 datasets, and there are detailed ablation studies to verify the effectiveness of each design. Experiments are repeated 5 times for reducing variance.
But for experiments about efficiency, I think there is still room for improvement. The authors only provide a comparison with RAP in terms of token usage, but I believe comparisons with other methods in Table 1 and comparison on inference time cost can be added to further prove the efficiency of this method.

**Methods And Evaluation Criteria:**

The method and the experiments are reasonable. Although controlling reasoning path via only one token embedding is a bit counter-intuitive, there are similar attempts in prior works like prompt tuning and other decoding strategies like FIRE and CoT-Decoding.

**Other Comments Or Suggestions:**

Please refer to "Claims and Evidence" and "Experimental Designs Or Analyses" sections for Weaknesses and Suggestions.

**Other Strengths And Weaknesses:**

Please refer to the above sections for Strengths and refer to "Claims and Evidence" and "Experimental Designs Or Analyses" sections for Weaknesses and Suggestions.

**Questions For Authors:**

1. It is intriguing that the authors have found that applying this approach can increase the activation rate of neurons by 3–4%, how do you understand this phenomenon, and have you tried to analyze why this occurs?

2. To my knowledge, there are methods trying to interpret "soft" tokens in the embedding space, e.g., finding their nearest neighbors (discrete tokens of LLMs) in the embedding space, have you tried to interpret the sampled first token in your framework?

**Relation To Broader Scientific Literature:**

The paper's approach is an exploration in the field of LLM's decoding strategies. It can be regarded as an extension to prior first-token based strategies like FIRE and CoT-Decoding, which is mentioned in the paper.

**Theoretical Claims:**

Most equations are seemingly correct to me.

---

> ### Author Rebuttal · Authors · 2025-04-01
>
> **1.Why Embedding Search Outperforms Discrete Sampling**
>
> > A theoretical provement of "searching for a start token in the continuous embedding space" being better than "sampling every token in the discrete token space" can further improve the soundness of the paper.
>
> Thank you for the suggestion. Searching for a start token in the continuous embedding space is preferable to sampling every token in the discrete token space from the theoretical, interpretability and the cost perspectives:
>
> - **Optimal solution to the objective.** The set of the embeddings of all tokens forms a discrete subset of our continuous space. Importantly, this means that the optimal solution (maximiser) to our objective function may **not** lie within this discrete subset.
> - **Role of the soft token.** As the soft token in our framework is not intended to approximate a specific discrete token, but rather to serve as a functional control token, influencing the overall response. Optimising in the continuous space allows exploration of representations that may lie between or beyond discrete tokens, enabling smoother control and better performance than fixed-token selection.
> - **High computational cost.** With the vocabulary size |V|>30k, sampling every token in the discrete token space requires generating the output for each token x and computing the corresponding f(x), making it computationally expensive at O(|V|).
>
>
> **2. Computational efficiency**
>
> > The authors only provide a comparison with RAP in terms of token usage, but I believe comparisons with other methods in Table 1 and comparison on inference time cost can be added to further prove the efficiency of this method.
>
> As the reviewer suggested, we additionally report inference time(min) cost across all baselines:
>
> |Method|GSM8K|GSM-Hard|SVAMP|StrategyQA|
> |-|-|-|-|-|
> |SC(τ=0.4)|26.58|33.91|20.54|20.04|
> |SC(τ=0.6)|26.12|34.46|21.76|19.87|
> |SC(τ=0.8)|27.28|34.86|21.16|20.80|
> |FIRE|26.70|32.26|21.59|20.17|
> |CoT-Decoding|26.56|32.55|21.53|20.60|
> |RAP|184.52|234.14|142.52|149.73|
> |Ours|**23.15**|**28.42**|**18.41**|**17.44**|
>
>
> Our method consistently achieves the lowest inference time across all tasks, further demonstrating its efficiency beyond token-level savings.
>
>
> **3. Neuron activation**
>
> > It is intriguing that the authors have found that applying this approach can increase the activation rate of neurons by 3–4%, how do you understand this phenomenon, and have you tried to analyze why this occurs?
>
> Thank you for the insightful question. We believe this phenomenon occurs because our perturbation strategy encourages broader exploration within the neuron space, thereby increasing the likelihood of activating previously dormant or marginal neurons across the architecture. In our setting, we define a neuron as `activated` if its activation value is greater than zero. As our perturbations produce more diverse token embeddings, the resulting activations tend to be more widely distributed across different neurons as well, thus increasing the overall number of activated neurons.
>
> However, we would like to stress that the overall increase in activation rate does not necessarily translate to improved final answer accuracy. Rather, the key contributing factor is the ability to activate more neurons identified as critical for the current problem instance [1]. The increased activation rate simply raises the likelihood of activating these critical neurons. The BO process then further optimises the soft embedding to consistently target them, as illustrated in Figure 4 of our paper.
>
> To further illustrate this point and to analyse the functional importance of these neurons, we have conducted an ablation study and found that **masking critical neurons** reduced accuracy from **62.14%** to **13.27%**, whereas random masking only dropped it to **41.89%**. This confirms that the activated neurons are non-random and impactful, consistent with prior findings on causal neurons [2].
>
> [1] Andy Zou et al., 2025. Representation Engineering: A Top-Down Approach to AI Transparency.
>
> [2] Kevin Meng et al., 2022. Locating and Editing Factual Associations in GPT. NeurIPS
>
>
>
> **4.Interpretation of the Soft Token**
>
> > Have you tried to interpret the sampled first token in your framework?
>
> We appreciate the reviewer’s suggestion. We conducted nearest-neighbor analysis to interpret the optimised soft token, but found it to be a clear outlier in the embedding space. Its Euclidean distance from all vocabulary tokens are several orders of magnitude beyond normal variation (e.g., z-score > 3000), suggesting that it does not correspond meaningfully to any real token. This is expected, as the soft token in our framework is not intended to approximate a specific discrete token, but rather to serve as a functional control token, influencing the overall response.

---

### Official Review · Reviewer_MeVe · 2025-03-15

**Overall Recommendation:** 3

**Summary:**

This paper proposed a Bayesian Optimization based approach to improve the test time performance of pre-trained LLMs. The authors propose to sample from an initially Gaussian distribution, where the perturbation vector is added to the distribution of the first token in the answer to control answer generation. In order to improve the answer, the authors proposed coherence reward and verifier reward as the objective for BO. Expected Improvement (EI) has been applied as the acquisition function.

**Claims And Evidence:**

Strength:
- The proposed algorithm improves the performance of three different base LLMs, and has demonstrated better performance than other perturbation/sampling algorithms.
- BO algorithm converges as shown in Figure 5.

Weaknesses:
- Lack of discussions and comparisons to other controlled generation works [1,2]

[1] Mudgal, Sidharth, et al. "Controlled Decoding from Language Models." International Conference on Machine Learning. PMLR, 2024.
[2] Qi, Xiangyu, et al. "Safety alignment should be made more than just a few tokens deep." arXiv preprint arXiv:2406.05946 (2024).

**Essential References Not Discussed:**

[1] Mudgal, Sidharth, et al. "Controlled Decoding from Language Models." International Conference on Machine Learning. PMLR, 2024.
[2] Qi, Xiangyu, et al. "Safety alignment should be made more than just a few tokens deep." arXiv preprint arXiv:2406.05946 (2024).

**Experimental Designs Or Analyses:**

Most experimental designs are fair, but I'd like to know more details about the choice and design of the verifier.

**Methods And Evaluation Criteria:**

There is one tricky part that makes me confused about the method:
- How is the verifier score calculated? What is the verifier? Is it another LLM?

In L189, there lacks of explanation about $y_v$. I think the choice of verifier matters a lot on the performance and has a large influence on the impact of this algorithm. If the verifier is another more performant LLM, I'd like to see the performance of that LLM. It is very hard to judge the contribution of this paper without such information and this also raises questions about whether the comparsion is fair.

Another question regarding algorithm is that:
- Why choose EI?

There exist multiple acquisition functions for BO, such as UCB score, etc., is there any theoratical/empirical results that support the choice of EI?

**Other Comments Or Suggestions:**

N/A

**Other Strengths And Weaknesses:**

See above.

**Questions For Authors:**

N/A

**Relation To Broader Scientific Literature:**

People working on autoregressive model might also be interested in this approach.

**Theoretical Claims:**

N/A

---

> ### Author Rebuttal · Authors · 2025-04-01
>
> **1. Verifier Setup**
>
> Regarding the questions about details of the verifier, **NO** separate or stronger LLM is used for verification (Line 31-33, right column); the same model as the generator is employed (i.e. LLaMA3-8B-Ins, Qwen2-7B-Ins, and Mistral-7B-Ins).
>
> - The verifier score $r_{verifier}(y) = \mathbb{1}_{y_v = y}$ is a binary indicator, where $ y_v$ is the model's regenerated answer based on the current question and previous responses. Specifically,  $y_v$ is obtained by prompting the same LLM with a verification query:
>
>   > *Based on the given question and the previous answers, please provide your analysis and final answer.*
>
>   The exact prompts used are listed in Appendix B.3.
> - We also conducted ablation studies (see *Verifier Comparison: Judgment vs. Generation*, Line 427, left column) to compare different verifier strategies and assess their impact on performance.
>
> This setup ensures a fair comparison, as no external or more capable model is used.  It also highlights one of the main contributions of this work: our BO framework's ability to enhance reasoning capabilities using a single unified LLM without additional verifiers. We will add these details on the setup of the verifier to our paper in the next revision.
>
> **2. Related Works**
>
> > Lack of discussions and comparisons to other controlled generation works [1,2]
>
> Thanks, we will add the comparison in the revision. Our goal is to propose an **efficient reasoning framework that does not require additional or stronger verifiers, nor task-specific model fine-tuning**. Instead, we leverage verification from the model itself to enhance the reasoning ability of LLMs.
>
> The methods in [1,2] serve different purposes: [1] uses trainable prefix scorers to guide decoding, while [2] proposes a fine-tuning objective aimed at improving robustness against adversarial prompts. Both approaches require additional training, making direct comparison with our training-free method less straightforward.
>
> We implemented a baseline based on [1] using a blockwise best-of-8 decoding (block size=16), where the model scores its own without any fine-tuning. To improve the scorer's quality, we applied self-consistency. For [2], we performed token-wise constrained fine-tuning on LLaMA3-8B-Ins model using LoRA (5 epochs, learning rate 2e-5, rank 16).
>
> |Method|Training|Shot|GSM8K|GSM-Hard|SVAMP|StrategyQA|
> |-|-|-|-|-|-|-|
> |Constrained Fine-tuning[2]|✅(lora)|-|78.3±0.7|13.6±0.5|83.5±0.7|**81.3**±0.8|
> |prefix scorer[1]|❌|Zero|75.2±0.9|26.1±1.3|83.6±0.9|65.2±1.3|
> |ours|❌|Zero|79.4±1.2|28.2±1.8|88.2±1.3|67.2±0.7|
> |prefix scorer[1]|❌|Few|81.2±1.6|33.6±1.4|88.5±1.2|72.4±1.1|
> |ours|❌|Few|**84.3**±1.4|**35.7**±1.0|**90.2**±0.6|75.6±0.8|
>
> As shown, while [2] achieves the best performance on the StrategyQA task with additional training, our method—without requiring any extra training—achieves superior results on the other three tasks, particularly in few-shot. In a fairer comparison (i.e. without training), our method consistently outperforms the prefix scorer [1] across all tasks.
>
> [1] Controlled Decoding from Language Models. ICML
>
> [2] Safety alignment should be made more than just a few tokens deep.
>
> **3. Why choose EI?**
> > There exist multiple acquisition functions for BO, such as UCB score, etc., is there any theoratical/empirical results that support the choice of EI?
>
> Indeed, PI and GP-UCB can also be used as the acquisition function. The cumulative regret for GP-UCB has the same rate (Theorem A.1) as EI [3]. Unlike EI or GP-UCB, PI only considers the probability of improvement, without accounting for its magnitude. It is thus less theoretically grounded and more prone to premature exploitation [4].
>
> |Method|GSM8K(ZS)|GSM8K(FS)|GSM-Hard(ZS)|GSM-Hard(FS)|SVAMP(ZS)|SVAMP(FS)|StrategyQA(ZS)|StrategyQA(FS)|
> |-|-|-|-|-|-|-|-|-|
> |ours(EI)|**79.4**±1.2|**84.3**±1.4|**28.2**±1.8|35.7±1.0|**88.2**±1.3|**90.2**±0.6|**67.2**±0.7|75.6±0.8|
> |PI|74.6±1.5|82.1±1.2|28.0±1.5|35.2±0.8|85.3±1.0|89.5±2.2|66.9±1.8|74.3±0.4|
> |UCB β=1|76.7±1.3|83.3±0.3|27.7±1.6|34.7±1.2|86.0±1.5|88.0±0.0|66.7±1.5|74.7±1.5|
> |UCB β=2|77.9±1.9|83.3±1.6|27.8±0.8|**36.2**±1.5|85.0±1.0|88.7±1.4|66.8±2.3|**75.7**±1.9|
> |UCB β=5|75.6±1.9|81.8±0.9|27.7±0.8|34.2±2.5|85.3±0.8|89.7±0.8|66.7±1.0|75.0±0.7|
>
>
> The experiments show that PI consistently underperforms compared to EI, as expected from the theoretical discussion above. For GP-UCB, its performance is sensitive to the choice of the exploration parameter and is, in most settings, worse than EI. We would also like to mention that the optimal parameter choice for GP-UCB varies across different tasks, making it difficult to guarantee good performance in unseen settings. In contrast, EI performs robustly without requiring task-specific tuning.
>
> [3] Taking the human out of the loop: A review of Bayesian optimization. Proc. IEEE
>
> [4] Gaussian process optimization in the bandit setting: No regret and experimental design. ICML

---

> > ### Comment · Reviewer_MeVe · 2025-04-04
> >
> > I appreciate the rebuttal from the authors. This has resolved my concern, so I would raise my score to weak accept.

---

> > > ### Author Response · Authors · 2025-04-08
> > >
> > > We truly appreciate Reviewer MeVe's thoughtful feedback. Your comments on the verifier design, choice of acquisition function, and related work were highly valuable in helping us refine our explanations and better contextualize our contributions. We will incorporate these improvements into the paper.

---

### Official Review · Reviewer_hpZB · 2025-03-15

**Overall Recommendation:** 4

**Summary:**

This work proposes a novel way of exploring the search space of LLM responses via perturbing the given input embedding in the generated sequence. In particular, authors design an online learning scheme that uses bayesian optimization to adjust parameters of the noise so that the generated outcome (after adjustment) leads to better reward or score. Authors conduct thorough comparison with strong baselines to show improvements coming out of from better search exploration in contrast to naive sampling with temperature.

**Claims And Evidence:**

- Simple token-level distribution adjustment (increasing the temperature) leads to higher chances of sampling hallucinated or degenerate outputs. The proposed method instead is steering the sequence in a desirable way according to a given reward/score model (or self-rewarding score)
- The idea of using random projections to reduce the dimension of the embedding to optimize is well motivated by highlighted issues of bayesian optimization in high-dimensional space

**Essential References Not Discussed:**

n/a

**Experimental Designs Or Analyses:**

I read experimental results and analysis: the choice of baselines makes sense. The only experiment which might be very useful here is to verify how useful this approach might be in RL training where we need to sample useful trajectories from an LLM. This method could be a promising direction for training-time sampling strategies.

**Methods And Evaluation Criteria:**

Proposed method effectively improves sample efficiency of the search within the search space when an outcome based reward or the score is available. Authors showed on real world examples how this allows us to find a better prediction or outcome compared to more usual best-of-n methods that iterate over independent samples from the model.

**Other Comments Or Suggestions:**

n/a

**Other Strengths And Weaknesses:**

In my opinion this work could get even more impact and recognition if authors might show the effectiveness of this approach in terms of sample efficiency when its used during training with preference optimization or reward-based training.

**Questions For Authors:**

n/a

**Relation To Broader Scientific Literature:**

The idea itself does not have any groundbreaking components in it, but its a smart application of known methods to come up with a very useful framework for guided generation with optimization during inference. Related work provides meaningful connection to other guided generation algorithms.

**Theoretical Claims:**

Methodology described in this work is clearly written, although I am not an expert in Bayesian optimization, in particular.

---

> ### Author Rebuttal · Authors · 2025-04-01
>
> > In my opinion this work could get even more impact and recognition if authors might show the effectiveness of this approach in terms of sample efficiency when its used during training with preference optimization or reward-based training.
>
> Thank you for this insightful and forward-looking suggestion. We agree that extending our embedding-based exploration method to training—particularly within preference optimisation frameworks—to enhance sample efficiency could be a promising direction.
>
> Specifically, our Bayesian optimisation approach to embedding perturbations can be integrated into preference optimisation as an efficient and targeted data generation mechanism. A key bottleneck in current reward-based training frameworks lies in the use of standard generation methods (e.g., temperature sampling), which often produce outputs that, while differing in perplexity, appear similarly plausible—making it difficult for reward models or human annotators to assign differentiated scores. Our proposed method could indeed offer an alternative approach. Instead of relying on random sampling or temperature-based decoding to generate candidate answers, our method explores promising regions in the embedding space, producing diverse and high-quality outputs with fewer trials. During training, we hope that these outputs can be evaluated using human preferences or reward models to construct preference pairs or ordered sequence with more significant distinct scores, which is able to provide more efficient supervision signals for optimising the model.
>
> If it is possible, by having those more distinctive generated samples and preference-aligned candidates in each iteration, the training process can access richer supervision without increasing the number of queries or annotations. This could lead to better utilisation of limited feedback and help the model learn more effectively from each batch of generated candidates. Additionally, one of the main properties of our proposed framework is that it is model-agnostic and lightweight. It can be seamlessly integrated into standard preference optimisation pipelines without modifying model architectures or requiring access to gradients.
>
> We appreciate the suggestion and view this integration as a promising approach for generating more preference-aligned data during training. We plan to explore its impact on training efficiency and reasoning performance in future work.

---

### Official Review · Reviewer_8Nqh · 2025-03-16

**Overall Recommendation:** 4

**Summary:**

This paper introduces a novel embedding-based search framework to enhance complex reasoning in Large Language Models (LLMs). It perturbs the embedding of the first token with Gaussian noise and optimizes this perturbation via Bayesian optimization (BO), guided by a verifier model. Experiments on multiple challenging datasets (GSM8K, GSM-Hard, SVAMP, StrategyQA) across three models (LLaMA, Qwen, Mistral) demonstrate consistent and significant accuracy improvements over existing strong baselines, especially in zero-shot settings. The approach achieves these gains efficiently, converging in few iterations without requiring access to model internals. Overall, the method offers a scalable and effective way to systematically improve LLM reasoning capabilities.

**Claims And Evidence:**

The claims made regarding improved accuracy, diversity of solutions, and computational efficiency are convincingly supported by thorough empirical evaluation. Results consistently favor the proposed method over baselines like Chain-of-Thought, self-consistency, and FIRE. One minor weakness is the claim about improving coherence, which is indirectly inferred from correctness but not explicitly measured.

**Essential References Not Discussed:**

The paper should discuss foundational work on **diverse decoding**, particularly "Diverse Beam Search" [1] (Vijayakumar et al., 2016), which explicitly balances diversity and quality in generation. Additionally, "Self-Refine" [2] (Madaan et al., 2023) is a relevant **iterative self-improvement** method, which aligns conceptually with the idea of model-guided refinement.

[1] Vijayakumar, A. K., Cogswell, M., Selvaraju, R. R., Sun, Q., Lee, S., Crandall, D., Batra, D. (2016). Diverse Beam Search: Decoding Diverse Solutions from Neural Sequence Models. Proceedings of the AAAI Conference on Artificial Intelligence, 30(1).

[2] Madaan, A., Yazdanbakhsh, A., Tandon, N., Gupta, P., Alon, U., Yang, Y., Yazdanbakhsh, A. (2023). Self-Refine: Iterative Refinement with Self-Feedback for Large Language Models. arXiv preprint arXiv:2303.17651.

**Experimental Designs Or Analyses:**

The experimental design is robust, sound, and fair. Results were validated with multiple seeds and clearly show low variance, enhancing reliability. Ablation studies (token placement, verifier strategies, embedding dimensionality) further validate design choices. A minor limitation is evaluating on subsets (200 samples per dataset), though this is mitigated by randomness and multiple seeds.

**Methods And Evaluation Criteria:**

The proposed methods (embedding perturbation and Bayesian optimization) are clearly justified and suitable for the problem of enhancing reasoning in LLMs. Evaluation criteria (accuracy, coverage), datasets (GSM8K, GSM-Hard, SVAMP, StrategyQA), and baselines used (CoT, FIRE, RAP, etc.) are appropriate and widely accepted for benchmarking LLM reasoning improvements.

**Other Comments Or Suggestions:**

1. **Verifier Reliability:** Could you clarify whether incorporating external or domain-specific verifiers (e.g., exact math calculators) is straightforward within your framework, and whether you expect substantial performance gains from a more reliable verifier?
2. **Scaling to Larger Models:** Have you performed initial tests on larger models (e.g., GPT-4 scale)? Would you anticipate similar performance improvements, or could diminishing returns become significant at larger scales?
3. **Optimization Scope:** Have preliminary experiments been conducted on optimizing embeddings for multiple tokens rather than just the first token, and would you expect significant gains from such extensions?

**Other Strengths And Weaknesses:**

**Strengths:**

- Strong empirical results demonstrating significant accuracy improvements.
- Computational efficiency and model-agnostic nature make it widely applicable.
- Clear and rigorous presentation of methods, analyses, and experiments.

**Weaknesses:**

- Dependence on potentially imperfect internal LLM verifier signals.
- Perturbation limited to the first token; may not correct deeper reasoning errors.
- Experiments restricted to smaller-scale models and specific reasoning tasks, raising questions about scalability to larger models or broader domains.

**Questions For Authors:**

1. **Verifier Reliability:** Could you clarify whether incorporating external or domain-specific verifiers (e.g., exact math calculators) is straightforward within your framework, and whether you expect substantial performance gains from a more reliable verifier?
2. **Scaling to Larger Models:** Have you performed initial tests on larger models (e.g., GPT-4 scale)? Would you anticipate similar performance improvements, or could diminishing returns become significant at larger scales?
3. **Optimization Scope:** Have preliminary experiments been conducted on optimizing embeddings for multiple tokens rather than just the first token, and would you expect significant gains from such extensions?

**Relation To Broader Scientific Literature:**

The paper advances beyond standard methods like chain-of-thought prompting and heuristic decoding strategies (self-consistency, FIRE, RAP) by proposing a principled embedding-space search strategy. Unlike discrete-token sampling methods, it leverages continuous embedding perturbation and Bayesian optimization, significantly improving reasoning accuracy. It also aligns with recent trends exploring LLMs' abilities for self-verification and iterative refinement.

**Theoretical Claims:**

The theoretical underpinnings (Bayesian optimization, embedding perturbation, dimension reduction) are sound and appropriately applied, though no fundamentally new theoretical insights are developed. The key theoretical novelty lies in the innovative combination of embedding-space optimization and BO for controlling LLM generation trajectories.

---

> ### Author Rebuttal · Authors · 2025-04-01
>
> **1. Verifier Reliability**
>
> Thanks for the comment. You are right in saying that incorporating more reliable, domain-specific verifiers could further improve performance. Our framework is flexible enough to integrate such tools, and we agree that doing so would likely yield gains in tasks where those verifiers are applicable [1].
> That said, our current focus is on improving reasoning in scenarios where such external tools are not available or applicable — for instance, in common-sense or social reasoning, where exact computation isn’t useful. By relying on internal LLM verifier signals, we aim to explore the model's capacity to self-reflect and reason without additional guidance  (Line 31-33, right column).
>
> [1] Large Language Models for Mathematical Reasoning: Progresses and Challenges. ACL
>
> **2. Scaling to Larger Models**
>
> Thanks for the suggestion. Note that our method involves modifications within the model’s parameter space, and due to the closed-source nature of GPT-4, it is not possible to test on it. Nonetheless, the point is well-taken and we have conducted additional experiments using Qwen2-72B-Instruct, a significantly larger model (72B) than those used in the main paper (7B and 8B), across both zero-shot and few-shot settings:
>
> |Method|AIME-2024(ZS)|AIME-2024(FS)|GSM8K(ZS)|GSM8K(FS)|GSM-Hard(ZS)|GSM-Hard(FS)|SVAMP(ZS)|SVAMP(FS)|StrategyQA(ZS)|StrategyQA(FS)|
> |-|-|-|-|-|-|-|-|-|-|-|
> |COT|0|3.3|91.0|91.0|51.5|65.0|93.0|92.0|79.0|90.0|
> |SC(τ=0.4)|3.3±2.7|3.3±3.3|93.3±0.6|91.8±1.0|62.3±0.6|68.7±1.5|93.2±0.3|93.8±0.3|78.2±0.8|89.6±1.4|
> |SC(τ=0.6)|2.2±3.3|2.2±3.8|93.7±0.3|92.2±1.0|62.7±0.3|68.2±1.4|93.8±0.6|93.1±0.5|78.8±1.6|90.0±1.3|
> |SC(τ=0.8)|3.3±1.9|2.2±1.9|94.0±0.5|93.5±0.5|62.8±2.0|68.3±0.3|93.7±0.3|93.7±0.3|78.4±2.5|88.8±2.3|
> |FIRE|2.2±1.9|3.3±3.8|91.4±0.8|92.5±0.4|60.3±0.6|65.7±2.0|93.5±0.9|93.5±0.5|78.2±1.9|89.5±0.9|
> |CoT-Decoding|2.2±1.9|2.2±1.9|93.6±1.9|93.5±1.1|61.0±2.3|66.0±1.7|**94.2**±1.2|93.8±1.5|78.8±1.6|89.0±1.5|
> |RAP|-|4.4±3.4|-|93.5±0.4|-|69.1±5.2|-|93.7±5.0|-|**90.4**±5.0|
> |Ours|**6.7**±2.7|**11.1**±1.7|**94.3**±0.3|**94.8**±1.3|**63.3**±0.6|**72.2**±0.6|94.0±1.0|**94.2**±1.3|**79.6**±0.3|89.2±1.2|
>
> While we still see some improvements, the gains are indeed smaller on easier benchmarks like GSM8K and SVAMP, as large models can already achieve very high accuracy.
> To further stress-test our method, we evaluated it on the more challenging `AIME-2024 dataset`. Here, we observe that our approach continues to yield significant gains even at the 72B scale, showing its robustness and scalability to harder tasks and more capable models. We will add these results to clarify the method’s applicability beyond smaller-scale settings.
>
>
> **3. Optimisation Scope**
>
> To investigate this, we conducted additional experiments where we optimised embeddings for the first k tokens (instead of just the first):
>
> |#token(k)|GSM8K(ZS)|GSM8K(FS)|GSM-Hard(ZS)|GSM-Hard(FS)|SVAMP(ZS)|SVAMP(FS)|StrategyQA(ZS)|StrategyQA(FS)|
> |-|-|-|-|-|-|-|-|-|
> |1(ours)|**79.4**±1.2|**84.3**±1.4|**28.2**±1.8|**35.7**±1.0|**88.2**±1.3|**90.2**±0.6|**67.2**±0.7|**75.6**±0.8|
> |2|75.0±1.8|83.3±1.3|24.8±0.6|34.7±1.0|83.0±0.5|90.2±1.4|68.5±0.5|74.2±1.3|
> |5|69.7±3.5|81.2±2.1|22.2±1.0|29.8±0.8|85.5±0.3|88.3±0.3|66.3±0.6|71.7±1.2|
> |10|61.0±3.1|73.7±2.3|17.2±1.6|23.8±0.3|82.8±0.8|86.8±0.8|67.3±1.9|71.7±1.6|
> |20|52.2±2.3|62.0±2.6|19.0±1.3|18.2±1.9|74.3±0.6|81.5±2.0|67.8±2.5|68.7±1.1|
>
> The performance generally degrades as k increases, especially beyond 5 tokens. This suggests that naively extending to multiple tokens can introduce instability or overfitting.
> We also compare with RAP (one of our baselines), a tree-search-based method that operates at the sequence level rather than token-by-token—though it shares similar ideas with token-by-token search. While RAP achieves strong performance, it incurs substantially higher cost and still underperforms our approach.
> Developing a multi-token optimisation strategy that can achieve both high accuracy and cost-effectiveness would require deeper investigation and extensive experimentation. It remains an interesting direction for future work and would warrant a separate paper.
>
> **4. Coherence Evaluation**
>
> We appreciate the reviewer’s observation. To address this, we evaluated coherence using two metrics: perplexity and a coherence score rated by DeepSeek-R1-Distill-Llama-70B, prompted to rate responses from 1 (poor) to 5 (excellent). We tested 800 samples (200 from each of four datasets) using LLaMA3-8B-Ins as the base model:
>
> |Type|SC(τ=0.4)|SC(τ=0.6)|SC(τ=0.8)|FIRE|CoT-Decoding|Ours|
> |-|-|-|-|-|-|-|
> |Perplexity ↓|7.22|8.51|9.98|6.87|6.73|**6.53**|
> |Coherence Score ↑|3.93|3.88|3.73|3.70|3.80|**4.08**|
>
> We would like to mention that coherence is not the main objective of our method, but serves as a filtering criterion to discard low-quality completions and support answer correctness.
>
> **5. Related Works**
>
> We thank the reviewer for the suggestions. We will discuss them in the final version.

---

> > ### Comment · Reviewer_8Nqh · 2025-04-03
> >
> > I appreciate the thorough response and the additional experiments that further strengthen the work. All of my concerns have been addressed, and I recommend accepting this submission.

---

> > > ### Author Response · Authors · 2025-04-08
> > >
> > > We sincerely thank Reviewer 8Nqh for your thoughtful feedback. Your comments have helped us improve the work, particularly through the additional experiments on scaling to larger models, multi-token optimisation, and coherence evaluation. We will incorporate the suggested clarifications and related work into the paper.

---

### Official Review · Reviewer_M1nu · 2025-03-17

**Overall Recommendation:** 3

**Summary:**

To address the challenges of insufficient diversity and low search efficiency in large language models (LLMs) for complex reasoning tasks, this paper introduces a novel responsive sampling strategy. By applying Gaussian perturbations to the embedding of the first token generated by the LLM, using the correctness and coherence of the output as the objective function, and leveraging Bayesian optimization to iteratively search for optimal embedding points, this approach avoids the blindness of traditional temperature tuning and the inefficiency of heuristic search. Experimental results demonstrate significant accuracy improvements on three mathematical reasoning datasets and one commonsense reasoning dataset. Additionally, the study validated that Bayesian iteration enhances neuron activation rates in the MLP layers of LLMs, providing neurophysiological evidence for the effectiveness of the proposed method.

**Claims And Evidence:**

The embedding optimization-based framework proposed in the paper significantly enhances the accuracy and efficiency of LLMs in complex reasoning tasks through Gaussian perturbation and Bayesian optimization. Experimental results demonstrate that it outperforms mainstream baseline methods on multiple benchmark datasets. Ablation studies validate the necessity of both the verifier and coherence terms, while neuron activation analysis reveals its mechanism of enhancing reasoning through diverse neural pathways and further proves the effectiveness of Bayesian optimization. A minor limitation is that the description of the verifier-guided approach remains somewhat ambiguous, particularly regarding how it produces a refined output y_v in the experimental section.

**Essential References Not Discussed:**

No

**Experimental Designs Or Analyses:**

In terms of dataset selection, recent research indicates that reasoning capabilities demonstrated in mathematical problems can generalize to other tasks. The paper also conducts experiments on more general commonsense reasoning tasks, validating the generalization ability of the proposed method. This dataset selection is therefore reasonable. For model selection, the paper employs Llama-3.1-8B-Instruct、Qwen2-7B-Instruct, and Mistral-8B Instruct as backbone models, avoiding dependence on specific architectures. Regarding baseline selection, the study includes CoT Prompting、Self-Consistency Decoding、FIRE、CoT-Decoding、multi-path generation and RAP (Monte Carlo Tree Search), covering both mainstream and state-of-the-art methods. The experimental section first evaluates the overall performance of the method on complex reasoning tasks. Under zero-shot and few-shot settings, it compares the proposed approach with strong mainstream baselines on three mathematical reasoning datasets and one commonsense reasoning dataset. Significant accuracy improvements demonstrate that controlled embedding exploration outperforms existing baselines in accuracy, diversity, and efficiency. The paper then conducts interpretability analysis by comparing neuron activation rates in MLP layers across different iterations, illustrating the effectiveness of Bayesian iteration. Additionally, ablation studies validate the contributions of the objective function and dimensionality reduction techniques.

**Methods And Evaluation Criteria:**

The paper proposes a responsive sampling method to address the challenges of insufficient answer diversity and low search efficiency in large language models (LLMs) for complex reasoning tasks. It ensures generation diversity through Gaussian embedding perturbation of the first token and greedy sampling, explores the embedding space via iterative Bayesian optimization to ensure accurate and coherent answers, and employs dimensionality reduction techniques to effectively tackle the high computational costs of high-dimensional embedding spaces. The method is compared with three strong baseline methods (CoT, Self-Consistency, FIRE) on three mathematical reasoning datasets and one commonsense reasoning dataset, validating its effectiveness in complex reasoning tasks.

**Other Comments Or Suggestions:**

No

**Other Strengths And Weaknesses:**

Strengths:
1. The main experimental results demonstrate significant improvements compared to baselines.
2. The methodology section is logically structured, guiding readers through the process of exploring the embedding space, Bayesian optimization, the required objective function and iterative procedures, and dimensionality reduction to mitigate inference costs. The appendices provide corresponding and detailed supplementary explanations.
3. The experimental design is rigorous, with sufficient motivation and extensive experimental validation for each component of the method.
Weaknesses:
1. While dimensionality reduction alleviates the curse of dimensionality, random projection may lose critical information. Further validation is needed to confirm that the negative impacts of dimensionality reduction are controllable.
2. Critical neurons play a pivotal role in the experiments, but their definition and identification rely solely on statistical-based approaches. The statistical significance of this method lacks theoretical justification, and it would be beneficial to supplement it with relevant prior work as a feasibility rationale.
3. Regarding computational efficiency, readers may expect comparisons of time and space costs rather than solely token count statistics.
4. It would be desirable to include additional details on the verifier’s experimental setup and implementation.

**Questions For Authors:**

No.

**Relation To Broader Scientific Literature:**

.

**Theoretical Claims:**

Yes, the theoretical foundation of the paper incorporates the Expected Improvement (EI) from Bayesian optimization theory and random projection methods for dimensionality reduction. These concepts are provided with specific and clear explanations in both the main text and the appendices.

---

> ### Author Rebuttal · Authors · 2025-04-01
>
> **1. Verifier Setup**
>
> Thanks for the comments. Regarding the questions about details of the verifier, **NO** separate or stronger LLM is used for verification (Line 31-33, right column); the same model as the generator is employed (i.e. LLaMA3-8B-Ins, Qwen2-7B-Ins, and Mistral-7B-Ins).
>
> - The verifier score $r_{verifier}(y) = \mathbb{1}_{y_v = y}$ is a binary indicator, where $ y_v$ is the model's regenerated answer based on the current question and previous responses. Specifically,  $y_v$ is obtained by prompting the same LLM with a verification query:
>
>   > *"Based on the given question and the previous answers, please provide your analysis and final answer."*
>
>   The exact prompts used are listed in Appendix B.3.
> - We also conducted ablation studies (see *Verifier Comparison: Judgment vs. Generation*, Line 427, left column) to compare different verifier strategies and assess their impact on performance.
>
> This setup ensures a fair comparison, as no external or more capable model is used.  It also highlights one of the main contributions of this work: our BO framework's ability to enhance reasoning capabilities using a single unified LLM without additional verifiers. Details on the verifier will be added to our paper.
>
> **2. Dimension Reduction**
> > While dimensionality reduction alleviates the curse of dimensionality, random projection may lose critical information. Further validation is needed to confirm that the negative impacts of dimensionality reduction are controllable.
>
> We agree that dimensionality reduction may lead to some information loss. However, it offers a practical tradeoff: retaining more information in high-dimensional spaces often results in poor sample efficiency for Bayesian optimisation, making it harder to find good configurations under a limited evaluation budget. As shown in Figure 6, we  experimented with several projection dimensions and found that our current setting (d=50) provides a good balance between optimisation performance and computational cost.
>
> To assess the stability of random projection, we ran simulations using 50 different random projection matrices. The box chart below shows the distribution of results:
>
> ```
> 0.850 ┤   ┐      ◀ Max
>       │   │
> 0.846 ┤ ┌─┘─┐    ◀ Q3
>       │ │   │
> 0.842 ┤ │ ─ │    ◀ Median
>       │ │   │
> 0.838 ┤ └─┐─┘    ◀ Q1
>       │   │
> 0.835 ┤   ┘      ◀ Min
> ```
>
> The performance remains stable across multiple runs, indicating that random projection does not introduce significant variance or instability.
>
> **3. Critical neurons**
>
> > The definition of critical neurons relies on statistical methods; citing related work would help justify this choice.
>
> We appreciate the reviewer’s concern. Prior works have shown that it is possible to trace information flow within transformers and identify neurons with causal influence on model predictions by applying targeted interventions, such as activation replacement or ablation [1, 2].
>
> Following this line of work, we **mask the critical neurons** identified for each input. This leads to a **significant drop** in accuracy to **13.27%** (from **62.14%** before masking). For comparison, we randomly masked the same number of neurons and repeated the experiment under identical settings, resulting in an average accuracy of **41.89%**. This substantial gap (**41.89%→13.27%**) demonstrates that the identified neurons are indeed functionally important, beyond what would be expected by chance.
>
> [1] Damai Dai et al., 2022. Knowledge Neurons in Pretrained Transformers. ACL
>
> [2] Kevin Meng et al., 2022. Locating and Editing Factual Associations in GPT. NeurIPS
>
> **4. Computational efficiency**
>
> > Regarding computational efficiency, readers may expect comparisons of time and space costs rather than solely token count statistics.
>
> We thank the reviewer for the suggestion. We supplement the token count statistics with comparisons between our method and RAP on inference time and memory usage. To evaluate the latter, we focus on the two variable components:
> (1) KV cache, and (2) intermediate activations, since model weights remain constant across methods. Using vLLM’s block-based memory tracking, we report both average and peak usage, sampled at 1-second intervals.
>
> |Dataset|Method|Time(min)|Intermediate Activations(avg,MB)|Intermediate Activations(peak,MB)|KV Cache(avg,MB)|KV Cache(peak,MB)|
> |-|-|-|-|-|-|-|
> |GSM8K|RAP|184.52|1628.7|1874.4|252.5|568.0|
> |GSM8K|Ours|**23.15**|**1137.2**|**1178.1**|**176.5**|**312.0**|
> |GSM-Hard|RAP|234.14|1985.4|2354.9|426.5|574.0|
> |GSM-Hard|Ours|**28.42**|**881.2**|**1096.2**|**254.6**|**336.0**|
> |SVAMP|RAP|142.52|1464.8|2089.5|384.5|494.0|
> |SVAMP|Ours|**18.41**|**932.4**|**1393.2**|**185.8**|**296.0**|
> |StrategyQA|RAP|149.73|1833.5|1935.9|241.4|376.0|
> |StrategyQA|Ours|**17.44**|**748.0**|**932.4**|**118.0**|**264.0**|
>
> The results show that our method's inference time is only 12.30% of RAP's while also consuming significantly less memory, further validating its computational efficiency advantage.

---

### Decision · Program_Chairs · 2025-05-01

**Decision:**

Accept (spotlight poster)

**Comment:**

All reviewers agree that the paper addresses a timely topic, employs a sound methodology, presents experiments that effectively support the claims, and is clearly written. The AC carefully checked the paper and agree with the reviewers.

Overall, this submission is strongly recommended for acceptance.